# Gradient Intrinsic Dimensionality Alignment: Bridging The Gap Between Low-Rank Adaptation and Full Fine-Tuning

**Jingqi Ye**[1,2*]   **Haonan He**[1,2*]   **Minglei Li**[1,3]   **Fujun Han**[1,4]   **Tao Chen**[3,6]   **Peng Ye**[1,5†]
[1]Shanghai Artificial Intelligence Laboratory   [2]University of Science and Technology of China
[3]Fudan University   [4]The Chinese University of Hong Kong, Shenzhen
[5]The Chinese University of Hong Kong   [6]Shanghai Innovation Institute
amzingye@mail.ustc.edu.cn

## Abstract

Parameter-Efficient Fine-Tuning (PEFT) techniques, such as Low-Rank Adaptation (LoRA) and its variants, have emerged as critical tools for adapting large pretrained models under limited computational resources. However, a notable performance gap persists between these LoRA methods and Full Fine-Tuning (FFT). In this paper, we investigate a key yet overlooked cause of this gap: the relationship between LoRA's low-rank adaptation subspace and true effective update directions of FFT gradients, which we define as the **gradient intrinsic dimensionality**. To systematically quantify this dimension, we first propose a novel entropy-based estimator, uncovering substantial discrepancies (up to more than 100x) between the rank of LoRA and the gradient intrinsic dimensionality. Motivated by this finding, we introduce **RaLoRA**, which adaptively aligns the ranks of LoRA adapters with layer-specific gradient intrinsic dimensions, without increasing the number of overall parameters. We further extend this approach into **RaLoRA-Pro**, integrating intra-layer rank alignment and inter-layer parameter reallocation guided by loss sensitivity, enabling finer-grained capacity relocation under comparable parameters. Extensive experiments demonstrate the effectiveness of our methods. Specifically, compared to vanilla LoRA, our methods achieve more than +5% improvement on GLUE, +0.57 on MT-Bench, +5.23% on GSM8K, +5.69% on HumanEval, and +1.58% on image classification, confirming consistent and substantial performance gains across diverse tasks and modalities.

## 1 Introduction

Large language models (LLMs) have shown remarkable generalization abilities across a broad range of downstream tasks (Yu et al., 2025; Wang et al., 2019; Adiwardana et al., 2020). However, full fine-tuning (FFT) LLMs on task-specific scenarios remains challenging due to the high GPU memory requirement. To address this challenge, various *Parameter-Efficient Fine-Tuning* (PEFT) methods (Houlsby et al., 2019; Hu et al., 2022; Li & Liang, 2021; Lester et al., 2021; Nguyen et al., 2023) have been proposed to adapt pretrained LLMs by updating a small number of parameters, hence reducing the memory requirement overhead. Among them, *Low-Rank Adaptation* (LoRA) (Hu et al., 2022) has become a widely adopted method due to its advantages, including strong empirical performance, zero inference latency, and easy implementation. LoRA approximates the weight update $\Delta W \in \mathbb{R}^{d_{out} \times d_{in}}$ using low-rank matrices $A \in \mathbb{R}^{r \times d_{in}}$ and $B \in \mathbb{R}^{d_{out} \times r}$, where $r \ll \min\{d_{in}, d_{out}\}$, thereby enabling efficient fine-tuning by updating low-rank matrices with constrained resources. Despite its efficiency and strong performance on simple tasks, LoRA still underperforms FFT on complex tasks such as mathematical reasoning (Biderman et al., 2024; Zhao et al., 2024).

Numerous variants of LoRA have been proposed aiming to bridge this performance gap, broadly categorized into three directions (Mao et al., 2025): (1) *Rank Augmentation*, which dynamically

---

*Equal contribution.
†Corresponding author: 20110720039@fudan.edu.cn

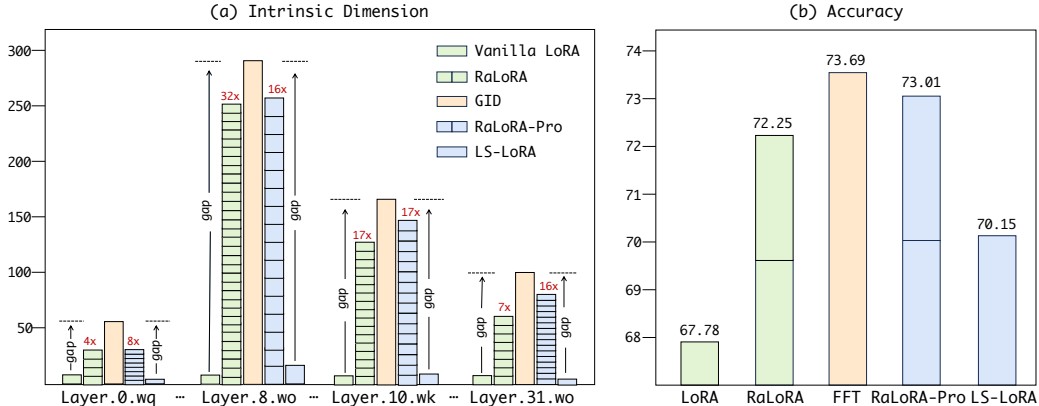

Figure 1: Intrinsic dimensionality and performance comparison across LoRA Variants and FFT on LLaMA3.1-8B-Base fine-tuned with MetaMathQA. GID denotes the gradient intrinsic dimensionality. LS-LoRA uses loss sensitivity-guided rank reallocation; RaLoRA and RaLoRA-Pro are our methods. Note that all methods except FFT have comparable numbers of trainable parameters.

reallocates the LoRA ranks across layers based on importance scores (He et al., 2025; Zhang et al., 2023a;b), or enhances expressiveness by stacking multiple LoRA modules (Ren et al., 2024; Lialin et al., 2023; Jiang et al., 2024); (2) *Optimization of Training Dynamics*, such as stabilizing the scaling factor (Kalajdzievski, 2023), assigning asymmetrical learning rates to $A$ and $B$ (Hayou et al., 2024), or decoupling the directional updates and magnitude updates (Liu et al., 2024a); and (3) *Improved Initialization*, including principal singular component-based (Meng et al., 2024) and orthonormal QR-based initialization strategies (Büyükakyüz, 2024), accelerating convergence and adaptation efficiency.

Nevertheless, existing approaches largely overlook a fundamental issue causing LoRA's performance shortfall: the misalignment between LoRA's low-rank update subspace and the true effective update directions of FFT gradients, which we define as the **gradient intrinsic dimensionality**. Our empirical investigation in Figure 1(a) reveals that the gradient intrinsic dimensionality in FFT typically occupies a significantly larger subspace (**up to 300**) than the fixed-rank structure permitted by LoRA (**usually set to 8**), potentially limiting its expressiveness, particularly on complex downstream tasks. Indeed, as illustrated in Figure 1(b), aligning LoRA's rank with the gradient intrinsic dimensionality substantially improves fine-tuning performance. However, realizing such alignment presents two primary challenges: (1) how to accurately estimate per-layer gradient intrinsic dimensionality—a largely unexplored issue; and (2) how to develop intrinsic dimensionality alignment strategies within fixed parameter budgets.

To address these challenges, we first propose a novel entropy-based estimator to quantify the gradient intrinsic dimensionality across layers systematically. This estimator is conceptually orthogonal to most existing LoRA variants and can even benefit other PEFT methods. Leveraging this estimator, we introduce **RaLoRA**, which utilizes block-diagonal decomposition (Ren et al., 2024) to dynamically align LoRA's ranks with the intrinsic dimensionality of the corresponding layer-wise gradients without increasing the number of trainable parameters. We further extend this approach to **RaLoRA-Pro**, which jointly optimizes intra-layer GID alignment and inter-layer parameter allocation guided by layer-wise loss sensitivity scores (Zhang et al., 2023b), offering more refined capacity relocation within comparable parameter budgets. Empirical results (Figure 1) confirm that our methods significantly reduce the performance gap between LoRA and FFT.

Our main contributions are summarized as follows:

1. We propose a novel entropy-based estimator that systematically quantifies the intrinsic dimensionality of gradients across layers. This estimator is conceptually orthogonal to most existing LoRA variants and can even benefit other PEFT methods.

2. We introduce **RaLoRA**, which adaptively aligns the rank of the LoRA module with the corresponding gradient intrinsic dimensionality without increasing the overall parameter count. We further present **RaLoRA-Pro**, which integrates GID alignment with loss sensitivity-guided parameter reallocation for fine-grained capacity control.

3. Extensive experiments demonstrate the effectiveness of our methods across diverse tasks and modalities. Specifically, compared to vanilla LoRA, our approach achieves over $5\%$ improvement on GLUE, $+0.57$ on MT-Bench, $+5.23\%$ on GSM8K, $+5.69\%$ on HumanEval, and $+1.58\%$ on image classification. These consistent results confirm that our methods can significantly narrow the performance gap to full fine-tuning.

## 2 RELATED WORKS

### 2.1 LoRA AND ITS VARIANTS

To efficiently adapt pretrained models under limited resources, PEFT methods such as LoRA (Hu et al., 2022) have gained prominence, though a noticeable performance gap remains compared to full fine-tuning. To bridge this gap, numerous variants of LoRA have emerged, which can be broadly categorized into three representative directions (Mao et al., 2025). (1) *Rank Augmentation*: Several methods aim to improve parameter utilization by dynamically reallocating the rank across layers based on importance scores (He et al., 2025; Zhang et al., 2023a;b). Others, such as MELoRA (Ren et al., 2024) and ReLoRA (Lialin et al., 2023), increase the equivalent rank by stacking multiple LoRA modules during fine-tuning. Furthermore, recent approaches approximate high-rank or full-rank updates through novel structural compositions, such as incorporating fixed random matrices (Albert et al., 2025) or leveraging the Hadamard product (Huang et al., 2025). (2) *Optimization of Training Dynamics*: This line of work focuses on improving training dynamics. For instance, RSLoRA (Kalajdzievski, 2023) stabilizes training by optimizing the scaling factor, while LoRA+(Hayou et al., 2024) assigns separate learning rates to the LoRA matrices $A$ and $B$ to stabilize the training process. DoRA (Liu et al., 2024a), in contrast, decouples the directional updates and magnitude updates to better align with task-specific gradients. (3) *Improved Initialization*: To accelerate convergence and improve adaptation, PiSSA (Meng et al., 2024) initializes LoRA modules with the principal singular components of the pretrained weight matrix, whereas OLoRA (Büyükakyüz, 2024) adopts orthonormal bases from QR decomposition for initialization, enhancing both stability and learning efficiency. While these methods offer improvements from various perspectives, they largely neglect a critical factor underlying the performance gap between LoRA and FFT: the misalignment between the low-rank update subspace of LoRA and the gradient intrinsic dimensionality.

### 2.2 EFFECTIVE RANK

The concept of effective rank was first introduced as a stable, continuous proxy for matrix rank based on the Shannon entropy of singular value distribution (Roy & Vetterli, 2007). Unlike a conventional defined rank, effective rank smoothly quantifies the intrinsic dimensionality and redundancy in data, making it particularly suitable for optimization tasks under uncertainty. Effective rank has found success in various domains. It has been applied in signal processing for dimensionality reduction and source separation (Deligiannidis & Doucet, 2019; ElMossallamy et al., 2020), and in machine learning for feature selection by capturing latent data complexity (Paul & Drineas, 2016). More recently, Yang et al. (Yang & Wang, 2023) used effective rank to assess the evolving quality of embeddings in Diffusion Probabilistic Models, while Nikitin et al. (Nikitin et al., 2024) employed it to distinguish between semantic diversity and true uncertainty in LLMs. Garrido et al. (Garrido et al., 2023) introduced RankMe, a self-supervised evaluation metric based on effective rank, effectively predicting downstream performance without labeled data. Inspired by its success across these domains, our work is the first attempt to introduce effective rank as a novel estimator for the intrinsic dimensionality of gradient matrices in the context of LoRA, a previously unexplored direction.

## 3 METHOD

### 3.1 BRIEF OVERVIEW OF LoRA

LoRA builds on the hypothesis that fine-tuning updates to pretrained weights lie in low-rank subspaces. It models the weight update $\Delta W \in \mathbb{R}^{d_{out} \times d_{in}}$ using low-rank matrices $A \in \mathbb{R}^{r \times d_{in}}$ and $B \in \mathbb{R}^{d_{out} \times r}$ as $\Delta W = \frac{\alpha}{r} BA$, while keeping the pretrained weight $W_0$ frozen. This design yields strong empirical performance with minimal memory requirement overhead, making LoRA one of the most widely adopted PEFT methods.

Theoretically, **LoRA adapter acts as an implicit gradient compressor**: at step $t$, it projects the full gradient $G_t$ onto a low rank subspace (see Appendix A for derivation):

$$\Delta(BA) \approx -\eta \left( B_t B_t^\top G_t + G_t A_t^\top A_t \right). \tag{1}$$

Consequently, the rank of LoRA fundamentally constrains both its gradient compression rate and its expressive power, which directly causes the performance gap observed relative to full fine-tuning.

### 3.2 Gradient Intrinsic Dimensionality Estimation via Entropy-Based Estimator

Building on the view that LoRA adapters act as gradient compressors, we introduce the notion of Gradient Intrinsic Dimensionality (GID) for the first time, which characterizes the true effective update directions of gradients of full fine-tuning. When GID substantially exceeds the preset rank $r$, such fixed-rank projections induce severe information loss and critically limit performance. Currently, most existing LoRA variants largely overlook this structural mismatch. This motivates us to develop a principled and adaptive framework that estimates GID and aligns LoRA ranks accordingly, thereby maximizing representational capacity under a fixed parameter budget. A straightforward approach to estimating the intrinsic dimensionality of $G$ is to apply singular value decomposition (SVD) and count the number of singular values above a small threshold $\varepsilon$:

$$rank(G) = \max\{i|\sigma_i > \varepsilon\}, \tag{2}$$

where $\sigma_1 > \sigma_2 > \cdots > \sigma_n$ are the singular values of the gradient matrix $G$. However, this approach is sensitive to the choice of $\varepsilon$, potentially limiting its robustness and generalizability when estimating layer-wise intrinsic dimensionalities across tasks.

To address these limitations, we introduce an entropy-based estimator inspired by the concept of effective rank (Roy & Vetterli, 2007; Su, 2025), which quantifies GID by examining the entropy of singular value distributions. Specifically, given the singular values $\sigma_1, \sigma_2, \ldots, \sigma_n$ of the gradient matrix $G_l$, the intrinsic dimensionality is defined as:

$$\text{erank}(G_l) = \exp\left( -\sum_{i=1}^{n} p_i \log p_i \right), \quad p_i = \frac{\sigma_i}{\sum_{j=1}^{n} \sigma_j}, \tag{3}$$

where $p_i$ denotes the normalized singular value distribution. See Appendix B for details.

Unlike threshold-based SVD methods that are sensitive to hyperparameters and lack adaptability across layers and tasks, our entropy-based estimator offers a **principled and robust measure** of the gradient matrices (see the empirical experiment in Appendix D). By quantifying the effective degrees of freedom based on the distribution of singular values, this method adaptively captures the underlying structure of gradients derived from full fine-tuning. This capability—previously underexplored in the context of LoRA—enables us to move beyond heuristic or importance-based rank assignment and instead align LoRA ranks with the actual gradient subspace complexity. As a general and lightweight module, our estimator not only underpins the design of RaLoRA and RaLoRA-Pro but can also be incorporated into broader LoRA-based or PEFT frameworks to guide adaptive GID alignment grounded in task-specific gradient behavior.

### 3.3 RaLoRA: Rank-Aligned LoRA with Gradient Intrinsic Dimensionality

We introduce RaLoRA, which generalizes LoRA by dynamically aligning its rank with the intrinsic dimensionality of each layer's gradient. As depicted in Figure 2(I), we adopt a structured parallel decomposition (Ren et al., 2024) that splits the LoRA matrices $A$ and $B$ into $n_l$ mini-blocks, forming a block-diagonal update matrix:

$$BA = \begin{pmatrix} B_1 A_1 & 0 & \cdots & 0 \\ 0 & B_2 A_2 & \cdots & 0 \\ \vdots & \vdots & \ddots & \vdots \\ 0 & 0 & \cdots & B_{n_l} A_{n_l} \end{pmatrix}.$$

where $A_i \in \mathbb{R}^{r \times (d_{\text{in}}/n_l)}$ and $B_i \in \mathbb{R}^{(d_{\text{out}}/n_l) \times r}$. The number of blocks $n_l$ for layer $l$ is determined by the estimated intrinsic dimension of its gradient $G_l$:

$$e_l = \left\lfloor \log_2\left( \frac{\text{erank}(G_l)}{r} \right) \right\rfloor, \quad n_l = 2^{e_l}, \quad \text{s.t.} \quad 1 \leq n_l \leq n_{\max}, \tag{4}$$

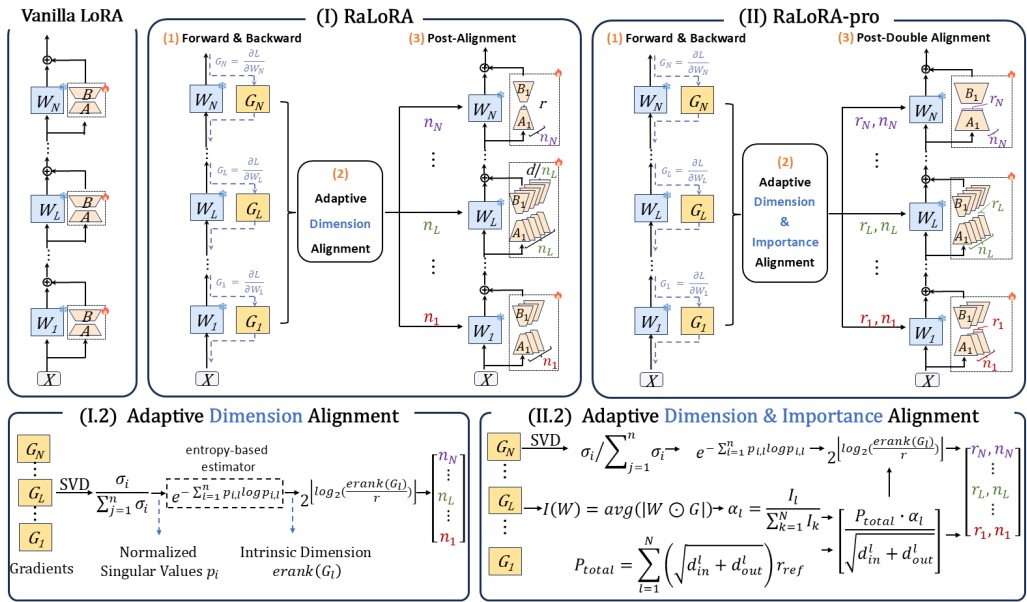

Figure 2: Overview of the proposed methods compared to vanilla LoRA. (I) RaLoRA adaptively aligns the rank of each LoRA adapter with the intrinsic dimensionality of full fine-tuning gradients estimated via an entropy-based estimator, using a block-diagonal decomposition while keeping the parameters fixed. (II) RaLoRA-Pro extends this by integrating a loss sensitivity-guided inter-layer reallocation strategy, achieving dual alignment under comparable parameters.

where $r$ is the vanilla LoRA rank, and the operator $\lfloor \cdot \rfloor$ ensures that $n_l$ is rounded down to the nearest power of 2, allowing it to evenly divide the input and output dimensions. The maximum expansion factor $n_{max}$ serves as an upper bound to maintain stability. This design **expands the equivalent rank to $n_l \times r$ while keeping the parameter count unchanged at** $r(d_{\mathbf{in}} + d_{\mathbf{out}})$.

Conceptually, **RaLoRA is a general extension of LoRA.** Guided by our entropy-based estimator, RaLoRA **adaptively adapts the number of blocks** $n_l$ per layer: when GID is low, it reduces to $n_l = 1$, identical to vanilla LoRA and focusing on dominant directions; when GID is high, it increases $n_l$, trading fine-grained precision for broader expressivity across multiple gradient subspaces.

This design is effective because **expressive power depends not only on parameter count but also on architectural structure.** By structurally aligning adaptation capacity with the GID, RaLoRA achieves more efficient use of parameters under the same budget, thereby improving approximation and task adaptability compared to vanilla LoRA. See theoretical analysis in Appendix C.

### 3.4 RaLoRA-Pro: Enhancing RaLoRA with Loss sensitivity-guided parameter reallocation

RaLoRA aligns the rank of LoRA with the intrinsic dimensionality of gradients at the intra-layer level. However, prior work has shown that the relative importance of different layers can vary significantly (He et al., 2025; Zhang et al., 2023b;a) given different models and tasks. Motivated by this observation, we propose RaLoRA-Pro as shown in Figure 2(II), which extends RaLoRA by integrating a loss sensitivity-guided inter-layer reallocation strategy (Zhang et al., 2022). This design enables **dual alignment** of LoRA adapters both intra layers (to match gradient intrinsic dimensionality) and inter layers (to reflect their relative importance), achieving finer-grained capacity distribution under a constrained parameter budget. Specifically, for a weight matrix $W_l \in \mathbb{R}^{d_{out} \times d_{in}}$ of the $l$-th layer, the importance score is defined as:

$$\mathrm{I}(W_l) = \mathrm{avg}(|W_l \odot G_l|), \tag{5}$$

where $\odot$ denotes element-wise multiplication. The importance score $\mathrm{I}(W_l)$ quantifies the average influence of layer parameters on the loss. To control the total number of trainable parameters, we

normalize the importance scores across $N$ target modules:

$$\alpha_l = \frac{I_l}{\sum_{k=1}^{N} I_k}, \tag{6}$$

where $\alpha_l \in [0, 1]$ indicates the relative importance of the $l$-th layer. This score guides proportional parameter allocation across layers. Assuming a reference rank $r_{\text{ref}}$, the total number of trainable parameters in vanilla LoRA with dimensionality smoothing [1] is given by:

$$P_{\text{total}} = \sum_{l=1}^{N} \left( \sqrt{d_{\text{in}}^l + d_{\text{out}}^l} \right) r_{\text{ref}}, \tag{7}$$

where $d_{\text{in}}^l$ and $d_{\text{out}}^l$ are the input and output dimensionalities of the $l$-th layer. Using the $\alpha_l$ calculated by the Equation 6, we reallocate the parameters (i.e., LoRA rank $r_l$) to each layer, ensuring that more influential layers receive higher adaptation capacity. The rank is computed as:

$$r_l = \left\lceil \frac{P_{\text{total}} \cdot \alpha_l}{\sqrt{d_{\text{in}}^l + d_{\text{out}}^l}} \right\rceil, \quad \text{s.t.} \quad r_{\min} \leq r_l \leq r_{\max}, \tag{8}$$

where $r_{\min}$ and $r_{\max}$ are predefined bounds that maintain stability. After parameter reallocation, we can align the dimensionalities of each layer according to section 3.3, achieving a dual alignment of FFT intra and inter layers. The full processes of RaLoRA-Pro are shown in Algorithm 1. Distinct from prior rank allocation methods that rely solely on sensitivity, RaLoRA-Pro represents the first dual-alignment framework that unifies inter-layer parameter reallocation with intra-layer geometric adaptation, ensuring that the allocated capacity structurally matches the effective update directions of full fine-tuning.

## 4 EXPERIMENTS

In this section, we present a comprehensive evaluation of RaLoRA and RaLoRA-Pro across natural language understanding (NLU), natural language generation (NLG), and image classification tasks. We compare against established baselines (see Appendix G.1), following experimental protocols from prior work (He et al., 2025; Wang et al., 2024b); full implementation details are provided in Appendix G. Results are organized as follows: Section 4.1 (NLU), Section 4.2 (NLG), and Section 4.3 (image classification). All experiments are repeated with three different random seeds.

### 4.1 EXPERIMENTS ON NATURAL LANGUAGE UNDERSTANDING

We fine-tune the T5-Base (Raffel et al., 2020) model on five subsets of the GLUE (Wang et al., 2018). Detailed training configurations are provided in Appendix G.3.

**Results.** As shown in Table 1, both RaLoRA and RaLoRA-Pro achieve the best overall performance among all LoRA variants, which corresponds to a substantial improvement of over $5\%$ compared to vanilla LoRA. Notably, on tasks such as QNLI and MRPC, our approach even surpasses FFT, despite using significantly fewer trainable parameters. These results suggest that aligning the rank of LoRA with the gradient intrinsic dimensionality enables more effective capacity utilization under small trainable parameter counts comparable to vanilla LoRA.

### 4.2 EXPERIMENTS ON NATURAL LANGUAGE GENERATION

We fine-tune LLaMA-3.1-8B-Base (Grattafiori et al., 2024) on three representative NLG tasks: math, code, and chat. Detailed training configurations are provided in Appendix G.4.

**Results.** As shown in Table 2, RaLoRA consistently outperforms all LoRA baselines across the three benchmarks with comparable trainable parameter counts. It achieves the best result on HumanEval, improving upon vanilla LoRA by +5.69, reducing the performance gap to FFT by $66.6\%$.

---

[1]The feature dimensionalities of different modules vary, which can introduce biases in importance estimation. Our dimensionality-smoothed allocation mitigates this issue, promoting fair adaptation across layers.

Table 1: Result of fine-tuning T5-Base using full fine-tuning and different LoRA variants on the five subsets of GLUE benchmark.

| Method | MNLI | SST-2 | CoLA | QNLI | MRPC | Average |
|---|---|---|---|---|---|---|
| Full | 86.33±0.00 | 94.75±0.21 | 80.70±0.24 | 93.19±0.22 | 84.56±0.73 | 87.91 |
| LoRA (Hu et al., 2022) | 85.30±0.04 | 94.04±0.11 | 69.35±0.05 | 92.96±0.09 | 68.38±0.01 | 82.08 |
| LoRA Variants with Rank Augmentation | | | | | | |
| MELoRA (Ren et al., 2024) | 85.57±0.19 | 93.62±0.30 | 77.28±0.36 | 89.59±5.24 | 84.11±1.45 | 86.03 |
| MoRA (Jiang et al., 2024) | 83.46±0.11 | 94.00±0.06 | 77.57±0.14 | 92.19±0.14 | 84.52±0.36 | 86.35 |
| AdaLoRA (Zhang et al., 2023b) | 85.45±0.11 | 93.69±0.20 | 69.16±0.24 | 91.66±0.05 | 68.14±0.28 | 81.62 |
| LoRA Variants with Optimized Training Dynamics | | | | | | |
| DoRA (Liu et al., 2024a) | 85.67±0.09 | 94.04±0.53 | 72.04±0.94 | 93.04±0.06 | 68.08±0.51 | 82.57 |
| RSLoRA (Kalajdzievski, 2023) | 85.73±0.10 | 94.19±0.23 | 72.32±1.12 | 93.12±0.09 | 52.86±2.27 | 79.64 |
| LoRA+ (Hayou et al., 2024) | 85.81±0.09 | 93.85±0.24 | 77.53±0.20 | 93.14±0.03 | 74.43±1.39 | 84.95 |
| LoRA Variants with Improved Initialization | | | | | | |
| PiSSA (Meng et al., 2024) | 85.75±0.07 | 94.07±0.06 | 74.27±0.39 | 93.15±0.14 | 76.31±0.51 | 84.71 |
| OLoRA (Büyükakyüz, 2024) | 85.39±0.17 | 92.85±0.05 | 76.89±0.54 | 92.73±0.23 | 83.75±0.92 | 86.32 |
| Ours | | | | | | |
| RaLoRA | 85.76±0.03 | **94.22±0.29** | **78.11±0.45** | 93.36±0.14 | 84.74±0.27 | **87.24** |
| RaLoRA-Pro | **85.82±0.13** | 94.00±0.17 | 77.85±0.35 | **93.38±0.04** | **85.09±0.39** | 87.23 |

On GSM8K, RaLoRA reaches 72.25, resulting in a +4.47 improvement over vanilla LoRA and recovering 75.6% of the FFT performance. The performance of RaLoRA on MT-Bench is also highly competitive, matching the strongest baseline (i.e., MoRA) while being significantly more stable. These results suggest that RaLoRA aligns the rank of LoRA with the corresponding GID, enhancing intra-layer expressiveness significantly without increasing the number of parameters.

Table 2: Performance of LLaMA-3.1-8B-Base fine-tuned with full fine-tuning and various LoRA variants on MT-Bench, GSM8K, and HumanEval. All methods use comparable trainable parameters, with LoRA rank and reference rank $r_{\text{ref}}$ in equation 7 set to 8.

| Method | MT-Bench | GSM8K | HumanEval |
|---|---|---|---|
| Full | 5.88±0.23 | 73.69±0.28 | 51.63±1.27 |
| LoRA (Hu et al., 2022) | 6.15±0.02 | 67.78±1.25 | 43.09±0.35 |
| MELoRA Ren et al. (2024) | 6.11±0.12 | 71.27±1.1 | 45.73±2.80 |
| MoRA (Jiang et al., 2024) | 6.38±0.12 | 68.12±0.91 | 29.68±5.73 |
| AdaLoRA (Zhang et al., 2023b) | 6.19±0.16 | 70.63±0.77 | 41.46±3.66 |
| DoRA (Liu et al., 2024a) | 6.24±0.12 | 69.17±1.00 | 43.70±1.54 |
| RSLoRA (Kalajdzievski, 2023) | 6.18±0.09 | 68.36±0.74 | 45.78±2.80 |
| LoRA+ (Hayou et al., 2024) | 6.35±0.10 | 71.29±0.93 | 44.51±2.11 |
| PiSSA (Meng et al., 2024) | 6.08±0.09 | 68.56±1.03 | 44.10±1.54 |
| OLoRA (Büyükakyüz, 2024) | 6.13±0.04 | 68.54±0.42 | 43.29±2.44 |
| RaLoRA(Ours) | 6.38±0.07 | 72.25±0.59 | **48.78±1.61** |
| RaLoRA-Pro(Ours) | **6.72±0.04** | **73.01±0.53** | 48.37±2.54 |

Beyond the strong performance of RaLoRA, RaLoRA-Pro achieves the best performance on MT-Bench and GSM8K, scoring 6.72 and 73.01, respectively. Compared to vanilla LoRA, these represent improvements of +0.57 and +5.23, effectively closing 88.5% of the performance gap with full fine-tuning on GSM8K. Importantly, RaLoRA-Pro even outperforms full fine-tuning on MT-Bench by +0.84. These findings suggest that the joint optimization of dual alignment in RaLoRA-Pro enables finer-grained control over adaptation capacity, providing a principled and effective PEFT strategy for complex tasks.

## 4.3 EXPERIMENTS ON IMAGE CLASSIFICATION TASKS

We fine-tune the CLIP-ViT-B/16 (Radford et al., 2021) model on seven widely used image classification benchmarks. Detailed training configurations are provided in Appendix G.5.

**Results.** As demonstrated in Table 3, both RaLoRA and RaLoRA-Pro significantly outperform existing LoRA variants. RaLoRA-Pro achieves the best result of 90.66%, a substantial gain of 1.58% over vanilla LoRA, while RaLoRA also shows a strong improvement of 1.45%. Notably, on challenging benchmarks such as Cars (fine-grained) and GTSRB (digit recognition), our methods surpass strong baselines such as OLoRA and MoRA, demonstrating robust performance across

Table 3: Results of CLIP-ViT-B/16 on image classification tasks using different LoRA variants. Note that zero-shot results are reported following (Wang et al., 2024c).

| Method | Cars | DTD | EuroSAT | GTSRB | RESISC45 | SUN397 | SVHN | Average |
|---|---|---|---|---|---|---|---|---|
| Zero-shot | 63.75 | 44.39 | 42.22 | 35.22 | 56.46 | 62.56 | 15.53 | 45.73 |
| LoRA (Hu et al., 2022) | 82.31±0.08 | 76.97±0.51 | 98.38±0.20 | 97.10±0.06 | 94.99±0.11 | 77.19±0.19 | 96.62±0.06 | 89.08±0.10 |
| **LoRA Variants with Rank Augmentation** | | | | | | | | |
| MELoRA (Ren et al., 2024) | 82.65±0.38 | 75.16±0.59 | 98.64±0.05 | 98.88±0.05 | 95.78±0.16 | 74.69±0.22 | 96.95±0.09 | 88.96±0.15 |
| MoRA (Jiang et al., 2024) | 84.61±0.21 | 77.34±0.14 | 98.65±0.16 | 98.68±0.18 | 96.33±0.19 | 78.12±0.06 | 97.17±0.15 | 90.13±0.16 |
| AdaLoRA (Zhang et al., 2023b) | 73.58±0.09 | 73.79±0.48 | 96.96±0.12 | 58.87±0.38 | 89.07±0.60 | 72.00±0.10 | 94.26±0.13 | 79.79±0.27 |
| **LoRA Variants with Optimized Training Dynamics** | | | | | | | | |
| DoRA (Liu et al., 2024a) | 82.44±0.26 | 76.86±0.84 | 98.43±0.17 | 97.25±0.12 | 95.10±0.16 | 77.30±0.17 | 96.63±0.04 | 89.14±0.07 |
| RSLoRA (Kalajdzievski, 2023) | 83.94±0.22 | 77.64±0.33 | 98.51±0.17 | 98.69±0.17 | 95.90±0.20 | 77.96±0.21 | 96.94±0.06 | 89.94±0.06 |
| LoRA+ (Hayou et al., 2024) | 86.61±0.36 | 73.33±1.30 | 98.54±0.14 | 98.99±0.20 | 96.06±0.38 | 76.80±0.34 | 96.98±0.08 | 89.62±0.19 |
| **LoRA Variants with Improved Initialization** | | | | | | | | |
| PiSSA (Meng et al., 2024) | 83.36±0.38 | 77.38±0.57 | 98.54±0.09 | 98.32±0.09 | 95.92±0.40 | 77.46±0.13 | 97.00±0.09 | 89.71±0.25 |
| OLoRA (Büyükakyüz, 2024) | 83.85±0.13 | 78.60±0.25 | 98.62±0.03 | 98.49±0.14 | 96.01±0.28 | 77.30±0.08 | 97.15±0.14 | 90.00±0.15 |
| **Ours** | | | | | | | | |
| RaLoRA | 86.63±0.30 | 77.75±0.20 | 98.66±0.27 | 98.98±0.11 | 96.62±0.28 | 77.86±0.05 | 97.24±0.11 | 90.53±0.03 |
| RaLoRA-Pro | 86.40±0.51 | 78.42±0.67 | 98.88±0.30 | 99.23±0.17 | 96.66±0.09 | 77.87±0.27 | 97.15±0.08 | **90.66±0.11** |

diverse visual domains. These results underscore that RaLoRA's alignment with the gradient intrinsic dimensionality and RaLoRA-Pro's dual alignment enable more efficient capacity allocation, yielding consistent improvements not only in NLP but also in vision tasks.

## 5 DISCUSSION

### 5.1 CHARACTERIZING GRADIENT INTRINSIC DIMENSIONALITY

We first quantify the gradient intrinsic dimensionality (GID) for each layer on Code-Feedback, MetaMathQA, and WizardLM datasets using our framework. As shown in Figure 4(a), Figure 3(a) and Figure 3(b), the estimated GID spans 30–1000, consistent with prior empirical findings that full fine-tuning gradients exhibit ranks 30–100× larger than standard LoRA settings (Biderman et al., 2024). Moreover, Mean GID correlates with task complexity: WizardLM ($\approx 404$), reflecting broad instruction diversity, exhibits the highest GID, followed by Code-Feedback ($\approx 269$) and MetaMathQA ($\approx 178$), in line with their respective representational demands.

We further track GID dynamics during fine-tuning on Code-Feedback, as shown in Figure 3(c) (see Appendix E.3 for implementation details). Across most layers, GID rises rapidly then stabilizes, highlighting both the robustness of our estimator and the persistence of dimensionalities far exceeding typical LoRA rank settings—an observation that may help explain the performance gap between LoRA and full fine-tuning.

Finally, Figures 3 reveal pronounced layer-wise heterogeneity in GID. Therefore, we further validate that our estimator captures such structure reliably, enabling finer-grained alignment strategies for parameter-efficient adaptation, see Appendix E.4 for experimental details.

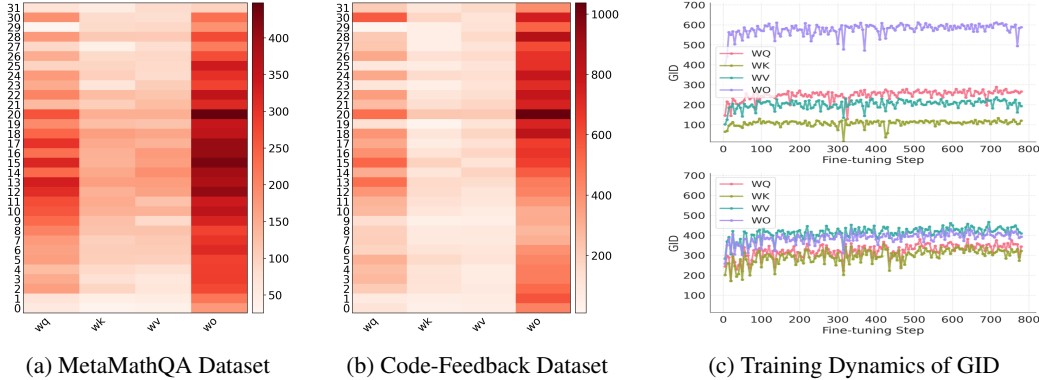

(a) MetaMathQA Dataset     (b) Code-Feedback Dataset     (c) Training Dynamics of GID

Figure 3: Layer-wise gradient intrinsic dimensionality (GID) across tasks and during training. (a,b): Heatmaps of estimated GID on MetaMathQA and Code-Feedback, obtained with our entropy-based estimator (WizardLM heatmap in Appendix 4a). (c): Dynamics of GID during Code-Feedback fine-tuning, showing rapid early growth followed by stabilization.

### 5.2 THE EFFECTIVENESS OF DIFFERENT STRATEGIES UNDER DIFFERENT RANKS

We conduct a controlled comparison of vanilla LoRA, LoRA only with loss sensitivity-guided rank reallocation (LS-LoRA), and our proposed methods RaLoRA and RaLoRA-Pro across different ranks, in order to isolate the contributions of intra-layer alignment and inter-layer reallocation under a comparable parameter budget.

As shown in Table 4, RaLoRA consistently outperforms vanilla LoRA across all tasks and rank settings, with notable improvements on GSM8K and HumanEval, further confirming the benefits of aligning rank with the gradient intrinsic dimensionality. Similarly, RaLoRA-Pro consistently surpasses LS-LoRA, especially at higher ranks, demonstrating the additional value of integrating intra and inter-layer alignment. Notably, LS-LoRA performs competitively on GSM8K at higher ranks but lags on MT-Bench and HumanEval—suggesting

Table 4: Performance comparison of LoRA, LS-LoRA, RaLoRA, and RaLoRA-Pro on MT-Bench, GSM8K, and HumanEval under different rank configurations.

| Rank | Method | MT-Bench | GSM8K | HumanEval |
|---|---|---|---|---|
| - | Full | 5.88±0.23 | 73.69±0.28 | 51.63±1.27 |
| 8 | LoRA | 6.15±0.02 | 67.78±1.25 | 43.09±0.35 |
| | RaLoRA | **6.19±0.20** | 71.42±0.87 | **47.76±1.27** |
| | LS-LoRA | 5.90±0.21 | 70.15±1.45 | 39.83±3.52 |
| | RaLoRA-Pro | 6.12±0.08 | **72.23±0.32** | 46.95±1.06 |
| 16 | LoRA | 6.16±0.12 | 67.83±0.81 | 42.68±0.61 |
| | RaLoRA | **6.28±0.02** | 72.12±0.79 | 48.37±3.57 |
| | LS-LoRA | 6.03±0.25 | 71.97±1.18 | 43.70±1.76 |
| | RaLoRA-Pro | 6.06±0.25 | **73.29±0.65** | **48.58±1.96** |
| 32 | LoRA | 6.14±0.14 | 68.01±1.14 | 42.68±1.22 |
| | RaLoRA | **6.22±0.14** | 73.37±1.35 | 48.98±2.75 |
| | LS-LoRA | 6.14±0.10 | 73.97±0.50 | 44.10±0.93 |
| | RaLoRA-Pro | 6.20±0.11 | **74.38±0.20** | **50.00±2.47** |
| 64 | LoRA | 6.28±0.09 | 67.17±0.61 | 43.29±2.20 |
| | RaLoRA | **6.32±0.06** | 74.55±0.46 | 51.22±2.11 |
| | LS-LoRA | 6.16±0.10 | 74.48±0.63 | 46.95±1.06 |
| | RaLoRA-Pro | 6.28±0.11 | **75.23±0.39** | **52.24±1.96** |

that inter-layer reallocation alone is insufficient for tasks requiring finer-grained adaptation. In contrast, RaLoRA-Pro's joint modeling of local gradient structure and global task importance provides consistently strong performance, highlighting the complementary value of combining intra- and inter-layer alignment. Interestingly, as the rank increases, our method achieves performance that surpasses full fine-tuning on these three tasks.

## 5.3 THE EFFECT OF MAX EXPANSION FACTOR

As shown in Table 5, we evaluate the impact of the maximum expansion factor $n_{\max}$ in Equation 4, which determines the number of parallel mini-LoRA blocks and thus the granularity of intra-layer adaptation. A larger $n_{\max}$ enables finer alignment with gradient intrinsic dimensionality but reduces parameter capacity per direction, creating a trade-off between alignment fidelity and expressiveness.

Our results show that the optimal $n_{\max}$ is task dependent. On GSM8K, performance peaks at $n_{\max} = 16$ and degrades with larger values, suggesting over-fragmentation harms tasks with lower intrinsic dimensionality. In contrast, on MT-Bench and HumanEval, RaLoRA's performance improves with larger $n_{\max}$, indicating that tasks with higher intrinsic dimensionality gradients benefit from finer alignment. These findings validate the necessity of introducing

Table 5: Effect of the maximum expansion factor $n_{\max}$ on the performance of RaLoRA and RaLoRA-Pro across MT-Bench, GSM8K, and HumanEval. mGID stands for mean gradient intrinsic dimensionality.

| $n_{\max}$ | Method | MT-Bench mGID $\approx$ 404 | GSM8K mGID $\approx$ 178 | HumanEval mGID $\approx$ 269 |
|---|---|---|---|---|
| 16 | RaLoRA | 6.18±0.12 | 72.25±0.59 | 45.12±1.22 |
| | RaLoRA-Pro | 5.75±0.08 | **73.01±0.53** | 48.37±2.54 |
| 32 | RaLoRA | 6.19±0.2 | 71.42±0.87 | 47.76±1.27 |
| | RaLoRA-Pro | 6.12±0.12 | 72.23±0.32 | 46.95±1.06 |
| 64 | RaLoRA | **6.38±0.07** | 71.19±0.79 | **48.78±1.61** |
| | RaLoRA-Pro | 6.05±0.11 | 71.24±0.5 | 45.73±2.66 |

$n_{\max}$ to control block granularity: when properly balanced with per-direction capacity, structural alignment with gradient dimensionality significantly improves adaptation performance.

## 5.4 THE EFFECT OF RANK REALLOCATION RANGE

Rank reallocation guided by loss sensitivity is a critical component of RaLoRA-Pro. To maintain parameter efficiency comparable to LoRA under a reference rank $r_{\mathrm{ref}}$, the reallocation range $(r_{\min}, r_{\max})$ in Equation 8 is typically chosen to satisfy the total parameter budget constraint. As shown in Table 6, we investigate the

Table 6: Performance comparison of different rank reallocation ranges in RaLoRA-Pro under a fixed reference rank $r_{\mathrm{ref}} = 8$ and maximum rank $n_{\max} = 32$.

| Rank range | MT-Bench | GSM8K | HumanEval | #Params |
|---|---|---|---|---|
| 8-8 | 6.19±0.2 | 71.42±0.87 | 47.76±1.27 | 6.82M |
| 4-16 | 6.12±0.08 | **72.23±0.32** | 46.95±1.06 | 6.32M |
| 4-32 | **6.70±0.23** | 71.39±1.02 | 45.32±1.76 | 7.00M |
| 6-32 | 6.65±0.30 | 71.37±0.88 | **48.17±0.61** | 7.84M |

impact of different rank allocation ranges under a fixed setting of $r_{\mathrm{ref}} = 8$ and $n_{\max} = 32$. For GSM8K, a relatively narrow range yields the best accuracy, suggesting that smaller deviations from

the reference rank may better preserve generalization in mathematical reasoning. On the other hand, broader ranges yield stronger performance on HumanEval and MT-Bench, indicating that tasks involving code generation and open-ended instruction following may benefit from more expressive and flexible adaptation. Overall, these findings highlight that appropriately selecting the rank reallocation range can yield substantial performance gains under comparable parameter budgets.

## 6 CONCLUSION

In this work, we address a fundamental and underexplored limitation of existing LoRA methods, the mismatch between LoRA and the intrinsic dimensionality of full fine-tuning gradients. To bridge this gap, we propose an entropy-based estimator that quantifies per-layer gradient intrinsic dimensionality in a robust and scalable manner. This estimator is orthogonal to most existing LoRA variants and can potentially serve as a plug-in tool for adaptive alignment with the gradient intrinsic dimensionality across a broad range of PEFT methods. Building on this insight, we introduce RaLoRA, which structurally aligns each LoRA adapter's rank with the estimated gradient intrinsic dimensionality. We further propose RaLoRA-Pro, which incorporates inter-layer importance to enable a dual-stage alignment mechanism. Extensive experiments across diverse NLP and image classification tasks show that our approaches significantly enhance LoRA performance under constrained parameter budgets and substantially narrow the performance gap with full fine-tuning. Considering the page limit, the discussion of this work's limitations and future works is deferred to the appendix I.

## ACKNOWLEDGEMENTS

This work was supported by a locally commissioned task from the Shanghai Municipal Government and the Shanghai Artificial Intelligence Laboratory (AILab). We also express our gratitude to our collaborators (Jingqi Ye, Haonan He, Minglei Li, and Fujun Han) for their valuable contributions during their internships at AILab, both in-person and remotely.

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

# A   DERIVATION OF LoRA AS A GRADIENT COMPRESSOR

Low-Rank Adaptation (LoRA) (Hu et al., 2022) posits that weight updates during fine-tuning reside in a low-rank subspace of the pretrained model's parameter space. Given a pretrained weight matrix $W_0 \in \mathbb{R}^{d_{\text{out}} \times d_{\text{in}}}$, LoRA decomposes the update $\Delta W \in \mathbb{R}^{d_{\text{out}} \times d_{\text{in}}}$ into low-rank matrices $A \in \mathbb{R}^{r \times d_{\text{in}}}$ and $B \in \mathbb{R}^{d_{\text{out}} \times r}$ as $\Delta W = \frac{\alpha}{r} BA$, while freezing $W_0$. Without loss of generality, we omit the scaling factor $\alpha/r$ (common in theoretical analyses), yielding the forward pass:

$$Y = W_0 X + BAX, \tag{9}$$

The gradients of the loss $\mathcal{L}$ with respect to $A$ and $B$ are derived via the chain rule. Let $G = \partial \mathcal{L} / \partial W$ represent the full gradient of the loss with respect to the weight matrix (i.e., $G = (\partial \mathcal{L} / \partial Y) X^\top \in \mathbb{R}^{d_{\text{out}} \times d_{\text{in}}}$, where $\partial \mathcal{L} / \partial Y$ is the upstream gradient). The gradients are:

$$\frac{\partial \mathcal{L}}{\partial A} = B^\top G, \tag{10}$$

$$\frac{\partial \mathcal{L}}{\partial B} = GA^\top. \tag{11}$$

Under stochastic gradient descent (SGD) with learning rate $\eta$, the updates for $A$ and $B$ are:

$$A_{t+1} = A_t - \eta \frac{\partial \mathcal{L}}{\partial A_t} = A_t - \eta B_t^\top G_t, \tag{12}$$

$$B_{t+1} = B_t - \eta \frac{\partial \mathcal{L}}{\partial B_t} = B_t - \eta G_t A_t^\top, \tag{13}$$

where $G_t = \partial \mathcal{L} / \partial W_t$ is the gradient at step $t$. The change in the low-rank update $\Delta(BA) = B_{t+1} A_{t+1} - B_t A_t$ is then:

$$\begin{aligned}
\Delta(BA) &= B_{t+1} A_{t+1} - B_t A_t \\
&= (B_t - \eta G_t A_t^\top)(A_t - \eta B_t^\top G_t) - B_t A_t \\
&= -\eta \left( B_t B_t^\top G_t + G_t A_t^\top A_t \right) + \eta^2 G_t A_t^\top B_t^\top G_t.
\end{aligned} \tag{14}$$

Since $\eta \ll 1$ in practice (typical values: $1e^{-5}$–$1e^{-3}$), the $\eta^2$-term is negligible. Thus, the first-order approximation is:

$$\Delta(BA) \approx -\eta \left( B_t B_t^\top G_t + G_t A_t^\top A_t \right). \tag{15}$$

This reveals that LoRA implicitly projects the full gradient $G_t$ onto a low-rank subspace spanned by $B_t B_t^\top$ and $A_t^\top A_t$. Crucially, when the true effective update directions of $G_t$ (termed the gradient intrinsic dimensionality of this paper) far exceed the LoRA rank $r$, this projection incurs significant information loss. Consequently, the expressivity of LoRA is fundamentally constrained by its rank $r$, necessitating a principled and adaptive gradient intrinsic dimensionality alignment framework to balance efficiency and performance.

# B   DETAILED EXPLANATION OF THE EFFECTIVE RANK ESTIMATION AND ITS PROPERTIES

## B.1   EFFECTIVE RANK DEFINITION

Consider a gradient matrix $G_l \in \mathbb{R}^{m \times n}$ at layer $l$, with its singular value decomposition (SVD) given by:

$$G_l = \sum_{i=1}^{\min(m,n)} \sigma_i u_i v_i^\top, \tag{16}$$

where $\sigma_1 \geq \sigma_2 \geq \cdots \geq \sigma_r > 0$ are the singular values, $r = \text{rank}(G_l)$, and $u_i \in \mathbb{R}^m$, $v_i \in \mathbb{R}^n$ are the corresponding left and right singular vectors.

Let $\sigma = (\sigma_1, \sigma_2, \ldots, \sigma_r)^\top$ denote the vector of non-zero singular values. Define the normalized singular value distribution as:

$$p_i = \frac{\sigma_i}{\|\sigma\|_1}, \quad i = 1, 2, \ldots, r, \tag{17}$$

where $\|\sigma\|_1 = \sum_{i=1}^r \sigma_i$ is the $\ell_1$-norm. The effective rank of $G_l$, denoted $\mathrm{erank}(G_l)$, is defined as:

$$\mathrm{erank}(G_l) = \exp(H(p_1, p_2, \ldots, p_r)), \tag{18}$$

where

$$H(p_1, p_2, \ldots, p_r) = -\sum_{i=1}^r p_i \log p_i \tag{19}$$

is the Shannon entropy of the distribution $\{p_i\}$. This entropy, also referred to as the *spectral entropy*, quantifies the spread of the normalized singular values and provides a robust measure of the gradient's intrinsic dimensionality for LoRA rank selection.

### B.2 PROPOSITION: BOUNDS ON THE EFFECTIVE RANK

The effective rank of a gradient matrix $G_l$ satisfies the following inequality:

$$1 \leq \mathrm{erank}(G_l) \leq \mathrm{rank}(G_l) \leq \min(m, n). \tag{20}$$

- The lower bound $\mathrm{erank}(G_l) = 1$ is achieved if and only if $\sigma = (\|\sigma\|_1, 0, \ldots, 0)^\top$, i.e., only one singular value is non-zero.

- The upper bound $\mathrm{erank}(G_l) = \mathrm{rank}(G_l)$ is achieved if and only if the non-zero singular values are all equal, i.e.,

$$\sigma = \left( \frac{\|\sigma\|_1}{r}, \ldots, \frac{\|\sigma\|_1}{r} \right)^\top.$$

**Proof.** The Shannon entropy of the normalized singular value distribution satisfies

$$0 = H(1, 0, \ldots, 0) \leq H(p_1, p_2, \ldots, p_r) \leq H\left(\tfrac{1}{r}, \ldots, \tfrac{1}{r}\right) = \log r. \tag{21}$$

Therefore,

$$1 \leq \mathrm{erank}(G_l) = \exp\left(H(p_1, \ldots, p_r)\right) \leq r. \tag{22}$$

Equality $\mathrm{erank}(G_l) = 1$ holds when $p_1 = 1$ and all others are zero, corresponding to $\sigma = (\|\sigma\|_1, 0, \ldots, 0)^\top$. Equality $\mathrm{erank}(G_l) = r$ holds when $p_i = 1/r$ for all $i$, meaning all non-zero singular values are equal.

## C ON THE EXPRESSIVE POWER OF RALORA: STRUCTURAL ALIGNMENT WITH GRADIENT INTRINSIC DIMENSIONALITY

### C.1 CORE ARGUMENT AND THEORETICAL REFINEMENT

We believe that **expressive power is not solely determined by parameter count but critically depends on architectural design** (He et al., 2016; Ye et al., 2025a; Yu et al., 2016; Ye et al., 2025b). Consequently, our central argument is that **RaLoRA enhances approximation quality by adaptively aligning with the gradient's intrinsic dimensionality (GID), thereby optimizing parameter utilization under a fixed budget**. This adaptive alignment enables RaLoRA to overcome limitations inherent in LoRA when gradients exhibit high-dimensional structure.

**RaLoRA is a generalized extension of LoRA.** Specifically, when the number of parallel blocks $n_l = 1$, RaLoRA reduces to LoRA, which provides **fine-grained approximation along dominant gradient directions**. As $n_l$ increases, RaLoRA adaptively **trades precision in dominant directions for broader expressivity**, capturing a more diverse spectrum of gradient subspaces. The selection of $n_l$ is adaptively driven by our proposed entropy-based estimator.

## C.2 Theoretical Analysis: Approximation Error Characterization

To characterize the approximation capabilities of LoRA and RaLoRA, we analyze their respective errors in representing the true weight update $\Delta W \in \mathbb{R}^{m \times n}$ under the Frobenius norm. Let the singular value decomposition (SVD) of $\Delta W$ be:

$$\Delta W = \sum_{i=1}^{\min(m,n)} \sigma_i u_i v_i^\top, \tag{23}$$

where $\sigma_1 \geq \sigma_2 \geq \cdots \geq 0$ are the singular values, and $u_i \in \mathbb{R}^m$, $v_i \in \mathbb{R}^n$ are the corresponding left and right singular vectors.

**LoRA's Approximation Error.** LoRA approximates $\Delta W$ with a rank-$r$ matrix $BA$ (ignoring scaling for clarity). By the Eckart–Young–Mirsky theorem, the optimal rank-$r$ approximation minimizes the Frobenius error, and the minimal error is:

$$E_{\text{LoRA}} = \min_{\text{rank}(BA) \leq r} \|\Delta W - BA\|_F^2 = \sum_{i=r+1}^{\min(m,n)} \sigma_i^2. \tag{24}$$

This implies LoRA captures only the top-$r$ singular directions. When the gradient exhibits slow spectral decay—i.e., high GID—the tail singular values $\sigma_{r+1}, \sigma_{r+2}, \dots$ remain significant, leading to substantial approximation loss.

**RaLoRA's block-diagonal structure Approximation Error.** RaLoRA approximates $\Delta W$ using a block-diagonal structure with $n_l$ parallel rank-$r$ blocks. We partition $\Delta W$ into an $n_l \times n_l$ block matrix $\{\Delta W_{p,q}\}_{p,q=1}^{n_l}$:

$$\Delta W = \begin{pmatrix} \Delta W_{1,1} & \Delta W_{1,2} & \cdots & \Delta W_{1,n_l} \\ \Delta W_{2,1} & \Delta W_{2,2} & \cdots & \Delta W_{2,n_l} \\ \vdots & \vdots & \ddots & \vdots \\ \Delta W_{n_l,1} & \Delta W_{n_l,2} & \cdots & \Delta W_{n_l,n_l} \end{pmatrix}, \tag{25}$$

Correspondingly, we partition each singular vector into $n_l$ segments:

$$u_i = \begin{pmatrix} u_{i,1} \\ u_{i,2} \\ \vdots \\ u_{i,n_l} \end{pmatrix}, \quad v_i = \begin{pmatrix} v_{i,1} \\ v_{i,2} \\ \vdots \\ v_{i,n_l} \end{pmatrix}, \quad \text{with } u_{i,p} \in \mathbb{R}^{m/n_l}, \ v_{i,q} \in \mathbb{R}^{n/n_l}. \tag{26}$$

Substituting Equation 23 into Equation 25, each block is:

$$\Delta W_{p,q} = \sum_{i=1}^{\min(m,n)} \sigma_i u_{i,p} v_{i,q}^\top. \tag{27}$$

RaLoRA approximates only the diagonal blocks ($p = q$) with low-rank matrices $B_p A_p$, while off-diagonal blocks ($p \neq q$) are implicitly set to zero. The total approximation error is therefore:

$$E_{\text{RaLoRA}} = \|\Delta W - \text{diag}(B_1 A_1, \dots, B_{n_l} A_{n_l})\|_F^2$$

$$= \underbrace{\sum_{p \neq q} \|\Delta W_{p,q}\|_F^2}_{\text{Truncation Error (off-diagonal blocks)}} + \underbrace{\sum_{p=1}^{n_l} \|\Delta W_{p,p} - B_p A_p\|_F^2}_{\text{Approximation Error (diagonal blocks)}}. \tag{28}$$

The truncation error arises from discarding off-diagonal interactions, while the approximation error reflects suboptimal low-rank fitting within each diagonal block. Crucially, **RaLoRA sacrifices fidelity to global top singular directions but gains by parallelizing learning across heterogeneous**

**subspaces**. This structural reorganization allows it to capture richer gradient information when GID is high, as each block specializes in distinct directional components.

To conclude, our analysis reveals that LoRA's performance is limited by its rigid rank, which struggles to approximate high-dimensional gradient updates. In contrast, RaLoRA's block-diagonal architecture and adaptive GID alignment, guided by an entropy-based estimator, enable it to effectively capture complex, high-GID. Consequently, RaLoRA provides a principled framework that aligns the adaptation structure with the GID, significantly enhancing the expressive power of low-rank adaptation.

## D    COMPARISON WITH THRESHOLD-BASED RANK ESTIMATION METHODS

To validate the robustness and superiority of our entropy-based estimator, we conduct a comparative study against conventional absolute threshold SVD methods for determining the gradient intrinsic dimension. The threshold-based approaches count the number of singular values $\sigma_i$ exceeding a specific tolerance $\varepsilon \in \{10^{-1}, 10^{-3}, 10^{-5}\})$. The estimated rank, $r_{\text{est}}$, is then converted to a block count $n_l$ using Eq. 4 . Results are averaged over 3 random seeds.

Table 7: Performance under different rank estimation strategies on GSM8K and HumanEval. "Ours" uses the entropy-based GID estimator. Threshold-based methods are highly sensitive to $\varepsilon$, while our approach achieves consistently superior performance without manual tuning.

| Task | Absolute Threshold ($\varepsilon$) | | | Ours |
|---|---|---|---|---|
| | $10^{-1}$ | $10^{-3}$ | $10^{-5}$ | |
| GSM8K | $67.15 \pm 0.50$ | $70.64 \pm 0.97$ | $69.30 \pm 1.32$ | $\mathbf{72.25 \pm 0.59}$ |
| HumanEval | $45.53 \pm 0.93$ | $44.71 \pm 1.27$ | $43.29 \pm 1.61$ | $\mathbf{48.78 \pm 1.61}$ |

**Analysis.**    The high sensitivity of threshold-based methods arises from a critical flaw: the spectral energy of gradients in large model fine-tuning decays slowly and non-uniformly, precluding a universal $\varepsilon$ for distinguishing "signal" from "noise." Heuristically setting a high threshold leads to capacity underutilization by aggressively truncating meaningful directions. Conversely, a low threshold retains optimization noise, diluting the signal-to-noise ratio and harming generalization. This instability is starkly demonstrated on HumanEval, where a performance drop of over 2.2 points is observed as $\varepsilon$ changes from $1e^{-1}$ to $1e^{-5}$.

In contrast, our entropy-based estimator offers a principled and robust alternative. It quantifies the true effective update directions of FFT gradients by measuring the entropy of singular value distributions. This data-driven, layer-adaptive approach naturally aligns with the underlying geometry of task-specific gradients, eliminating the need for manual tuning. This capability, previously underexplored in LoRA, allows us to move beyond heuristic rank assignment and directly align LoRA ranks with the actual gradient subspace complexity.

## E    DETAILS FOR CHARACTERIZING GRADIENT INTRINSIC DIMENSIONALITY

### E.1    GID ACROSS NLG TASKS AND LAYERS

To gain deeper insights into how gradient intrinsic dimensionality varies across different tasks and network layers, we employ our entropy-based estimator to analyze the full fine-tuning gradients of LLaMA-3.1-8B-Base on three representative datasets: Code-Feedback, MetaMathQA, and WizardLM. Figure 4 reveal distinct, non-random functional patterns regarding intra-layer roles and inter-layer depth. These patterns offer insights into how LLMs adapt to varying task demands.

**Intra-layer Functional Variation ($W_o$ Dominance).**    We observe substantial heterogeneity across attention sub-modules. Specifically, the output projection weight $W_o$ consistently exhibits significantly higher GID compared to $W_q$, $W_k$, and $W_v$. This high intrinsic dimensionality suggests that $W_o$ serves a critical role in aggregating and synthesizing information from diverse attention heads, requiring a higher-rank update subspace to effectively transform context-dependent features.

**Task-Specific Depth Dependency.** The distribution of GID across the network depth varies by task, reflecting distinct adaptation requirements:

- **Open-ended Generation (Code-Feedback, WizardLM):** These tasks display high GID in the upper layers. This indicates that upper layers in Transformers are responsible for high-level semantic abstraction and output formatting, which are critical for code generation and complex instruction following.

- **Logical Reasoning (MetaMathQA):** Conversely, mathematical reasoning tasks exhibit higher GID concentrations in the middle layers. This suggests that the adaptation required for logical reasoning relies heavily on transforming intermediate representations rather than surface-level realization.

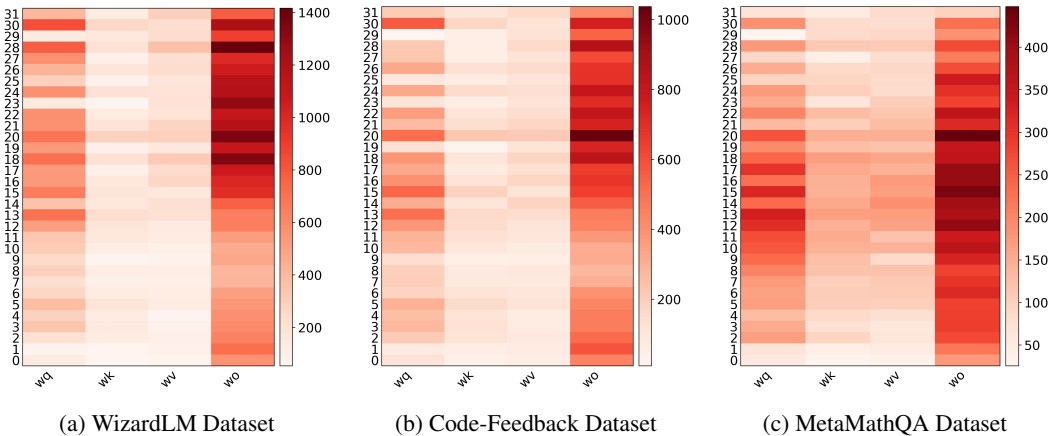

    (a) WizardLM Dataset         (b) Code-Feedback Dataset         (c) MetaMathQA Dataset

Figure 4: Heatmap of layer-wise gradient intrinsic dimensionality of LLaMA-3.1-8B-Base on Code-Feedback, MetaMathQA, and WizardLM Datasets, Illustrating task-specific and intra-layer variations in representational complexity.

### E.2    CORRELATION WITH FISHER INFORMATION AND LOSS CURVATURE

To determine if GID is merely a proxy for loss sensitivity, we analyzed its relationship with the Fisher Information (FI) matrix. As established in prior work (Kunstner et al., 2019), the FI matrix approximates the Hessian, serving as an effective measure of second-order local curvature.

The diagonal of the empirical Fisher Information for model parameters $\theta$ on dataset $\mathcal{D}$ is defined as (Kim et al., 2025):

$$\mathcal{F}_\theta(\mathcal{D}) \approx \frac{1}{|\mathcal{D}|} \sum_{x \in \mathcal{D}} \left( \frac{\partial \mathcal{L}_{\mathrm{LM}}(x; \theta)}{\partial \theta} \right)^2 \tag{29}$$

Intuitively, $\mathcal{F}_\theta$ captures the *sensitivity* or *importance* of parameters regarding the loss landscape. We denote the average FI value for a layer $l$ as $\bar{\mathcal{F}}_l$.

**Correlation Analysis.** We computed the Pearson and Spearman correlation coefficients between the layer-wise estimated GID and the layer-wise Fisher Information $\bar{\mathcal{F}}_l$. The results, averaged across tasks, are presented in Table 8.

Table 8: Correlation analysis between GID and FI ($\mathcal{F}_l$) across different module types. The low correlation values indicate that GID captures properties distinct from parameter sensitivity.

| Module | Pearson ($\log \mathcal{F}_l$) | Pearson ($1/\mathcal{F}_l$) | Spearman Rank |
|--------|--------|--------|--------|
| $W_q$ | -0.051 | -0.261 | 0.125 |
| $W_k$ | -0.179 | 0.141 | -0.113 |
| $W_v$ | -0.172 | -0.214 | -0.053 |
| $W_o$ | 0.277 | -0.251 | 0.325 |

Our empirical results demonstrate **no significant correlation between GID and Fisher Information**. While initially counterintuitive, this finding provides a profound insight into the optimization geometry of PEFT and theoretically justifies the design of our methods.

**Magnitude $\neq$ Dimension.** The lack of correlation indicates that *sensitivity* (how much the loss changes, measured by FI) and *structural complexity* (how many directions are needed for the update, measured by GID) are **decoupled properties**.

- A layer can be highly sensitive (steep loss landscape, high FI) but structurally simple, requiring updates along only a few dominant principal directions (low GID).

- Conversely, a layer may require processing diverse, fine-grained features necessitating a high-rank update (high GID), even if it is not the most dominant factor in the overall loss magnitude (moderate FI).

**Justification for RaLoRA-Pro.** This "orthogonality" explicitly validates the dual-alignment strategy employed in RaLoRA-Pro. Since GID and parameter importance (Loss Sensitivity/FI) capture distinct aspects of the optimization landscape, an effective PEFT method must address both simultaneously:

1. **RaLoRA** aligns ranks with GID to match the *intra-layer structural complexity*.
2. **RaLoRA-Pro** further integrates Loss Sensitivity to match the *inter-layer parameter importance*.

If GID and FI were highly correlated, the dual-alignment strategy would be redundant. The observed lack of correlation explains why RaLoRA-Pro—which considers both dimensions—consistently outperforms methods that rely principally on importance-based allocation (e.g., AdaLoRA (Zhang et al., 2023b)) or fixed-rank structures.

### E.3  GRADIENT INTRINSIC DIMENSIONALITY DYNAMICS DURING FINE-TUNING

We tracked the gradient intrinsic dimensionality (GID) in section 5.1 using the entropy-based estimator for each layer every 5 update steps under the following configuration:

- **Model:** Llama-3.1-8b-Base
- **Dataset:** Code-Feedback
- **Optimizer:** Adalomo(Lv et al., 2023a;b) (layerwise SGD with adaptive learning rate, memory efficient and can be easily used to conduct analysis)
- **Batch size:** 8 (per GPU) $\times$ 8 GPUs = 64
- **Total update steps:** 1024
- **Learning rate:** 1e-4 (common choice for Adalomo optimizer)

The first and last layers throughout training, visualized in Figure 3(c). Key observations include:

- **Dynamic Evolution of GID:** The GID exhibits a rapid increase during the early stages of training ($\leq 20$ steps) before stabilizing. The subsequent stabilization confirms that the gradient subspaces converge to high-dimensional manifolds that far exceed the ranks of standard LoRA.

- **GID Magnitude vs. LoRA Rank:** We find that the estimated GIDs consistently exceed typical LoRA ranks (e.g., $r = 8$–$64$). This suggests that LoRA's fixed low-rank bottleneck discards critical gradient information, which is a key reason for its performance gap with full fine-tuning.

- **Layer-wise Heterogeneity:** The GID varies significantly across different layers, with some layers consistently maintaining a higher intrinsic dimensionality than others. This heterogeneity provides the motivation for RaLoRA's adaptive, layer-specific rank allocation, which aligns LoRA capacity with the true complexity of per-layer GID.

Table 9: Performance and parameter efficiency comparison on GSM8K and HumanEval. LoRA+GID assigns per-layer ranks based on estimated GID, matching parameter counts of standard LoRA for fair comparison.

| | GSM8K | | | HumanEval | |
|---|---|---|---|---|---|
| Method | Accuracy (%) | Params | Method | Pass@1 (%) | Params |
| FFT | $73.69 \pm 0.28$ | 1.34B | FFT | $51.63 \pm 1.27$ | 1.34B |
| LoRA (r=8) | $67.78 \pm 1.25$ | 6.82M | LoRA (r=8) | $43.09 \pm 0.35$ | 6.82M |
| LoRA (r=200) | $74.12 \pm 1.23$ | 170M | LoRA (r=380) | $49.39 \pm 2.66$ | 323M |
| LoRA+GID | $\mathbf{75.41 \pm 0.84}$ | 175M | LoRA+GID | $\mathbf{52.64 \pm 1.27}$ | 323M |

### E.4 EVALUATING ORTHOGONALITY AND FINE-GRAINED ALIGNMENT OF THE ENTROPY-BASED ESTIMATOR

We designed a set of comparative experiments to concurrently assess our estimator's orthogonality with LoRA like methods and its reliability in capturing layer-wise heterogeneity. To this end, we set the LoRA rank for each layer to the GID estimated by our estimator (denoted as **LoRA+GID**). For a fair comparison, we increased the rank of standard LoRA so that its number of trainable parameters was as close as possible to that of LoRA+GID. We conducted these experiments on both mathematics and code generation tasks, keeping the implementation details consistent with those in Section 4.2. We also included performance comparisons against full fine-tuning and a standard LoRA with a fixed rank of 8 to provide a more comprehensive and intuitive set of results.

Results are shown in Table 9. When the rank of LoRA is increased, the performance improves significantly, which demonstrates that LoRA's low-rank bottleneck restricts its performance. In addition, LoRA+GID still achieves significant performance improvements when the number of trainable parameters is comparable to LoRA. This is because the proposed entropy-based estimator has good orthogonality and can effectively estimate the GID of each layer to identify the heterogeneity between different layers, thereby having a finer-grained alignment capability.

## F ALGORITHM

The full processes of RaLoRA-Pro are shown in Algorithm 1.

## G EXPERIMENT DETAILS

### G.1 BRIEF OVERVIEW OF BASELINE METHODS

We compare RaLoRA and RaLoRA-Pro with several relevant or widely used methods:

1. **Full Fine-Tuning**: Updates all parameters in designated layers. This method serves as the upper bound in terms of performance but incurs the highest memory and computation cost.

2. **LoRA** (Hu et al., 2022): Introduces a pair of low-rank matrices $A$ and $B$ per fine-tuned layer. Only these matrices are updated during fine-tuning, significantly reducing the number of trainable parameters.

3. **LoRA Variants with Rank Augmentation:** These methods aim to increase the rank of LoRA while maintaining a fixed parameter budget:
   - **AdaLoRA** (Zhang et al., 2023b): Prunes less important updates based on SVD and importance scores, reallocating rank to more influential layers within a constrained budget.
   - **MELoRA** (Ren et al., 2024): Employs multiple parallel mini-LoRA modules to enhance equivalent rank during training.
   - **MoRA** (Jiang et al., 2024): Utilizes non-parametric square matrices to implicitly achieve high-rank updates via structured transformations.

---

**Algorithm 1** The RaLoRA-Pro algorithm

---

**Input:** Model $f(\cdot)$ with target FT weight $W_l, l = 1, \ldots, L$, sample mini-batch size $N$, loss $\mathcal{L}$,
  reference rank $r_{ref}$, max blocks $n_{\max}$

**Output:** Initialized LoRA matrices $\{A_{l,i}, B_{l,i}\}$       ▷ $l = 1, \ldots, L$ and $i = 1, \ldots, n_l$

  1: **for** $l = 1$ to $L$ **do**
  2:      $G_l^{avg} \leftarrow 0$
  3:      $(d_{out}, d_{in}) \leftarrow \text{size}(W_l)$
  4:      $P_{total} = \left(\sqrt{d_{in} + d_{out}}\right) r_{ref}$               ▷ Pre-define trainable parameters budget.
  5: **for** $i = 1$ to $N$ **do**
  6:      Sample mini-batch $\mathcal{B}_i = \{x, y\}$, compute $\hat{y} \leftarrow f(x)$
  7:      $\ell \leftarrow \mathcal{L}(y, \hat{y})$
  8:      **for** $l = 1$ to $L$ **do**
  9:         $G_l^{avg} += \frac{1}{N}\frac{\partial \ell}{\partial W_l}$    ▷ Accumulate gradients of pretrained weights without optimizer step.
10: **for** $l = 1$ to $L$ **do**
11:      $I(W_l) \leftarrow \text{avg}(|W_l \odot G_l^{avg}|)$
12: Normalize importance: $\alpha_l = \frac{I_l}{\sum_{k=1}^N I_k}$
13: **for** $l = 1$ to $L$ **do**
14:      $(d_{out}, d_{in}) \leftarrow \text{size}(W_l)$
15:      $r_l \leftarrow \text{clip}(\text{round}(P_{total}\alpha_l/\sqrt{d_{in} + d_{out}}), r_{\min}, r_{\max})$
16:      Compute $\text{erank}(G_l^{avg})$, set $n_l \leftarrow 2^{\text{clip}(\lfloor \log_2(\text{erank}/r_l)\rfloor, 1, n_{\max})}$
17:      Partition $A_l, B_l$ into $n_l$ blocks: $A_{l,i}, B_{l,i}$
18:      **for** $i = 1$ to $n_l$ **do**               ▷ Initialize weights of the adapter.
19:         $A_{l,i} \leftarrow \text{Kaiming init}$
20:         $B_{l,i} \leftarrow \text{Zero init}$
21: **Return** $\{A_{l,i}, B_{l,i}\}$               ▷ $l = 1, \ldots, L$ and $i = 1, \ldots, n_l$

---

4. **LoRA Variants with Optimized Training Dynamics:** These methods have been proposed to enhance the optimization process during fine-tuning:

- **DoRA** (Liu et al., 2024a): Decomposes pretrained weights into magnitude and direction, updating only the direction component using LoRA for efficient adaptation.
- **RSLoRA** (Kalajdzievski, 2023): Optimizing the scaling factor to stabilize training.
- **LoRA+** (Hayou et al., 2024): Assigning different learning rates to the LoRA matrices $A$ and $B$.

5. **LoRA Variants with Improved Initialization:** These approaches aim to enhance LoRA performance through improved initialization strategies.

- **OLoRA** (Büyükakyüz, 2024): Initializing LoRA matrices with orthonormal bases from QR decomposition to improve convergence and stability.
- **PiSSA** (Meng et al., 2024): Using principal components from SVD of the pretrained weights to initialize LoRA modules for more effective adaptation.

### G.2 HYPERPARAMETER SETTINGS FOR BASELINE METHODS AND OUR METHODS

In the baselines of this article, there are several adjustable hyperparameters introduced by LoRA variants. We detail the hyperparameter configurations for all compared LoRA variants. Unless otherwise noted, we follow the recommended settings reported in the original papers to ensure a fair comparison.

- **LoRA** (Hu et al., 2022): Unless specified, we set the LoRA rank $r = 8$ (for RaLoRA-Pro, this refers to the reference rank $r_{\text{ref}}$), and the scaling factor $\alpha = 16$. The matrix $A$ is initialized using Kaiming initialization, while matrix $B$ is initialized to zero.
- **LoRA+** (Hayou et al., 2024): The learning rate ratio between matrices $A$ and $B$ is set to 16.

- **AdaLoRA** (Zhang et al., 2023b): The initial rank is set to 12 and the target rank to 8, with $t_i = 150$ and $t_f = 900$.

- **PISSA** (Meng et al., 2024): The number of SVD iterations is set to 64.

- **MELoRA** (Ren et al., 2024): We set $n = 2$ and split only the input dimensionality of matrix $A$ and the output dimensionality of matrix $B$, keeping the rank dimensionality fixed.

- **MoRA** (Jiang et al., 2024): We apply a rotation-based compression and reconstruction scheme.

- **RaLoRA (Ours)**: Unless specified otherwise, the number of parallel blocks is set to $n_{\max} = 32$.

- **RaLoRA-Pro (Ours)**: Given a reference rank $r_{\text{ref}}$, we set $r_{\min} = r_{\text{ref}}/2$, $r_{\max} = 2r_{\text{ref}}$, and $n_{\max} = 32$ by default.

### G.3 TRAINING CONFIGURATIONS FOR NATURAL LANGUAGE UNDERSTANDING

**Models and Datasets.** We fine-tune the T5-Base (Raffel et al., 2020) model on five subsets of the GLUE (Wang et al., 2018) benchmark: MNLI, SST-2, CoLA, QNLI, and MRPC. Performance is reported on validation sets using accuracy as the metric.

**Implementation Details.** We use the Adam (Kingma & Ba, 2014) optimizer with $\beta_1 = 0.9$, $\beta_2 = 0.999$, weight_decay=0. The batch size is set to 32, and a cosine learning rate schedule with a warmup ratio of 0.03 is applied. All linear layers except the language modeling head are fine-tuned using FP32 precision. The peak learning rate is set to $1e-4$, and the maximum input sequence length is 128. Please refer to Appendix G.2 for the detailed hyperparameter settings of different LoRA variants.

### G.4 TRAINING CONFIGURATIONS FOR NATURAL LANGUAGE GENERATION

**Models and Datasets.** We fine-tune LLaMA-3.1-8B-Base (Grattafiori et al., 2024) on three representative NLG tasks: math, code, and chat. Evaluation is conducted as follows:

**1.** *Mathematical reasoning*: Trained on a 100K subset of MetaMathQA (Yu et al., 2023) and evaluated on GSM8K (Cobbe et al., 2021). Accuracy is computed using regex-based extraction.
**2.** *Code generation*: Trained on a 100K subset of Code-Feedback (Zheng et al., 2024) with code-only labels and evaluated on HumanEval (Chen et al., 2021) using PASS@1 as the metric.
**3.** *Conversational dialogue*: Trained on a 52K subset of WizardLM (Xu et al., 2024). Evaluation follows the MT-Bench (Zheng et al., 2023) single-turn dialogues evaluation protocol, reporting average scores (0–10) from GPT-4o, Gemini-1.5-Pro, and LLaMA-3.1-70B-Instruct, following the standardized prompts of (Zheng et al., 2023) for comparability.

**Implementation Details.** We use the AdamW (Loshchilov & Hutter, 2017) optimizer with $\beta_1 = 0.9$, $\beta_2 = 0.999$, weight_decay $= 5e-4$. The batch size is 64. A cosine decay learning rate schedule with a decay rate of 0.1 and a warmup ratio of 0.03 is used. The peak learning rate is set to $5e-4$ for AdaLoRA, and $5e-5$ for full fine-tuning and all other baselines. Fine-tuning is restricted to the attention projection matrices: $W_q$, $W_k$, $W_v$, and $W_o$. For RaLoRA, we search over the set $\{16, 32, 64\}$ for the expansion factor $n_{\max}$ and report the best-performing configuration, which is further discussed in Section 5.3. For RaLoRA-Pro, a grid search strategy is used, and the best result is reported. Specifically, we sweep the maximum expansion factor $n_{\max} \in \{16, 32, 64\}$ and evaluate three different rank ranges: $(r_{\min}, r_{\max}) \in \{(4, 16), (4, 32), (6, 32)\}$.

### G.5 TRAINING CONFIGURATIONS FOR IMAGE CLASSIFICATION TASKS

**Models and Datasets.** We fine-tune the CLIP-ViT-B/16 (Radford et al., 2021) model on seven widely used benchmarks: StanfordCars (Krause et al., 2013), DTD (Cimpoi et al., 2013), EuroSAT (Helber et al., 2019), GTSRB (Houben et al., 2013), RESISC45 (Cheng et al., 2017), SUN397 (Xiao et al., 2010), and SVHN (Netzer et al., 2011). Model performance is reported on the validation sets using accuracy as the evaluation metric.

**Implementation Details.** During fine-tuning, only the linear layers of the vision backbone are updated, while all other parameters remain frozen. Training is performed in FP32 precision. For classification, we adopt prompt-based templates such as "a photo of a {class}." The Adam optimizer is employed with hyperparameters $\beta_1 = 0.9$, $\beta_2 = 0.999$, weight decay = 0, and a batch size of 64. We use a cosine learning rate schedule with 3% warmup and a peak learning rate of 1e-4. Zero-shot results are taken from (Wang et al., 2024c), and all fine-tuned results are averaged over three different random seeds.

## H    EXTENDED COMPARATIVE ANALYSIS AND DISCUSSION

To provide a comprehensive assessment of our contributions within the broader PEFT landscape, this section extends our evaluation beyond standard LoRA variants. We present comparative analyses against non-LoRA approaches (Section H.1) and alternative high-rank adaptation strategies (Section H.2), followed by a discussion on recent redundancy-reduction methods (Section H.3).

### H.1    COMPARISON WITH NON-LORA PEFT ARCHITECTURES

We compare RaLoRA and RaLoRA-Pro against representative non-LoRA methods that employ distinct adaptation mechanisms:

- **(IA)$^3$** (Liu et al., 2022): Adapts the model by learning vectors to rescale inner activations (Key, Value, and FFN), prioritizing extreme parameter efficiency.
- **FourierFT** (Gao et al., 2024): Learns sparse updates in the spectral domain via the Discrete Fourier Transform, theoretically enabling efficient fine-tuning of large models.

**Experimental Results.** We evaluated these methods on the GSM8K and HumanEval benchmarks following the protocols in Section 4.2. As detailed in Table 10, both RaLoRA and RaLoRA-Pro consistently outperform (IA)$^3$ and FourierFT. Specifically on GSM8K, RaLoRA-Pro achieves a substantial gain of $+19.54\%$ over FourierFT and $+6.39\%$ over (IA)$^3$. These results suggest that structurally aligning adaptation rank with gradient intrinsic dimensionality offers significantly higher expressivity than vector rescaling or spectral updates.

Table 10: Performance comparison with non-LoRA PEFT methods on GSM8K and HumanEval. "$-$" indicates failure to converge.

| Method | GSM8K | HumanEval |
|---|---|---|
| (IA)$^3$ (Liu et al., 2022) | $66.62 \pm 1.01$ | $-$ |
| FourierFT (Gao et al., 2024) | $52.71 \pm 4.69$ | $-$ |
| LoRA (Hu et al., 2022) | $67.78 \pm 1.25$ | $43.09 \pm 0.35$ |
| **RaLoRA (Ours)** | $72.25 \pm 0.59$ | $\mathbf{48.78 \pm 1.61}$ |
| **RaLoRA-Pro (Ours)** | $\mathbf{73.01 \pm 0.53}$ | $48.37 \pm 2.54$ |

**Analysis of Convergence on Complex Tasks.** A critical observation is the failure of both (IA)$^3$ and FourierFT to converge when fine-tuning on the Code-Feedback dataset. We attribute this optimization failure to the task's high complexity. As illustrated in Figure 3b of the main paper, the Code-Feedback dataset exhibits a significantly higher mGID ($\approx 269$) compared to standard instruction tuning. We hypothesize that the restricted adaptation subspaces of (IA)$^3$ and FourierFT lack the sufficient capacity to capture these high-dimensional gradient updates. In contrast, RaLoRA adaptively expands rank capacity to match this complexity, ensuring robust convergence and superior performance.

**Implementation Details.** For fairness, we utilized the official implementations from the PEFT library with hyperparameters aligned with their respective original publications: (IA)$^3$ used a learning rate of $3e^{-3}$ adapting $\{W_q, W_k, W_v, W_o, W_{\text{up}}\}$; FourierFT used a learning rate of $3e^{-2}$, `n_frequency`=1000, and scaling factor 300.

## H.2 COMPARISON WITH HIGH-RANK LoRA VARIANTS

We further compare our approach against HiRA (Huang et al., 2025), a representative method that utilizes the Hadamard product to achieve high-rank updates without increasing the parameter budget.

**Theoretical Distinction.** While HiRA effectively increases rank, it relies on a heuristic Hadamard product mechanism that does not guarantee alignment with the underlying optimization landscape. In contrast, RaLoRA is grounded in the theoretical view of the adapter as a gradient compressor (Appendix A). By employing our entropy-based estimator, RaLoRA explicitly identifies the GID of FFT gradients and constructs a block-diagonal structure to match the GID. Thus, RaLoRA represents a *data-driven* adaptive alignment strategy, whereas HiRA remains *structurally heuristic*.

**Empirical Results.** We compared the methods on GSM8K and HumanEval using standard settings for HiRA ($lr = 1e^{-3}$). As shown in Table 11, both RaLoRA and RaLoRA-Pro outperform HiRA across both benchmarks, validating the efficacy of GID-guided alignment.

Table 11: Performance comparison between HiRA and our methods. All methods utilize a comparable parameter budget (equivalent to rank $r = 8$).

| Method | GSM8K | HumanEval |
|---|---|---|
| LoRA ($r = 8$) | $67.78 \pm 1.25$ | $43.09 \pm 0.35$ |
| HiRA ($r = 8$) | $69.93 \pm 0.39$ | $45.73 \pm 1.22$ |
| **RaLoRA (Ours)** | $72.25 \pm 0.59$ | $\mathbf{48.78 \pm 1.61}$ |
| **RaLoRA-Pro (Ours)** | $\mathbf{73.01 \pm 0.53}$ | $48.37 \pm 2.54$ |

## H.3 DISCUSSION ON RECENT PEFT ADVANCEMENTS

Finally, we clarify the distinction between our contributions and other recent works.

MLAE (Wang et al., 2024a) and FVAE-LoRA (Kumar et al., 2025) primarily address redundancy within LoRA adapters. MLAE utilizes masking strategies to enforce independence among experts, while FVAE-LoRA employs a variational autoencoder framework to disentangle task-relevant information. While effective at reducing parameter redundancy, these methods do not explicitly measure or align with the GID.

Unlike the aforementioned approaches, our work is the first to explicitly quantify the dimensional gap between LoRA and FFT using a novel entropy-based estimator. By aligning the adapter's rank with the estimated GID, we ensure that the adaptation capacity structurally matches the task-specific optimization landscape.

## I  LIMITATIONS AND FUTURE WORKS

This paper aims to address a central yet previously unexplored issue in LoRA: the mismatch between LoRA's low-rank adaptation subspace and the gradient intrinsic dimensionality. To quantify this mismatch, we introduce an entropy-based estimator to systematically estimate the intrinsic dimensionality of gradient matrices arising from full fine-tuning. Although our empirical analysis demonstrates that the entropy-based estimator successfully captures meaningful differences in intrinsic dimensionality across layers and tasks, the effectiveness of this estimator currently remains empirical and requires extensive theoretical analysis or broader empirical validation.

Moreover, the entropy-based estimator proposed in this work is conceptually orthogonal to most existing LoRA variants (Kalajdzievski, 2023; Hayou et al., 2024; Wang et al., 2024b; Meng et al., 2024; Büyükakyüz, 2024; Li et al., 2026) and even to other PEFT approaches (Zhao et al., 2024). It thus holds promise as a general-purpose tool to facilitate adaptive rank selection based on the intrinsic gradient dimensionality of task-specific full fine-tuning, potentially narrowing the performance gap between parameter-efficient methods and full fine-tuning. However, in this study, we have only explored three specific applications (RaLoRA, RaLoRA-Pro and LoRA+GID in Appendix E.4).

Investigating how to effectively integrate our entropy-based estimator with other existing LoRA variants and PEFT approaches remains an important avenue for future research.

Finally, given the rapid advancement of multimodal large language models (MLLMs), which have become a pivotal area in artificial intelligence, it is crucial to evaluate the proposed method on various multimodal benchmarks to assess its generalizability and effectiveness. In future work, we plan to extend our framework to multimodal settings and evaluate its performance on comprehensive benchmarks such as (Liu et al., 2024b; 2023; Han et al.; 2024).

## J  LLM USAGE DISCLOSURE

During the preparation of this manuscript, we employed a large language model solely for linguistic refinement, including addressing potential grammatical issues and enhancing clarity and readability. The model was not used to generate scientific content, ideas, experiments, or analyses. The authors take full responsibility for the accuracy and integrity of the manuscript.

