# OpenReview forum: "Gradient Intrinsic Dimensionality Alignment：Narrowing The Gap Between Low-Rank Adaptation and Full Fine-Tuning"
_ICLR.cc/2026/Conference — ICLR 2026 Poster_

### Official Review · Reviewer_4tTg · 2025-10-30

**Soundness:** 3
**Presentation:** 3
**Contribution:** 3
**Rating:** 6
**Confidence:** 4

**Summary:**

This paper introduces a novel and compelling concept: the mismatch between the fixed, low-rank subspace of LoRA and the high intrinsic dimensionality (GID) of full fine-tuning gradients is a fundamental limitation of existing LoRA variants. To address this, the authors propose an entropy-based estimator for Gradient Intrinsic Dimensionality (GID) and build two new methods upon it: RaLoRA, which performs intra-layer rank alignment, and RaLoRA-Pro, which adds inter-layer parameter reallocation. The paper is well-structured, the motivation is clear and well-argued, and the experimental evaluation across NLP and vision tasks is extensive and generally demonstrates the superiority of the proposed methods.

**Strengths:**

Novel and Well-Motivated Core Idea: The paper's central thesis—that aligning LoRA's structure with the intrinsic dimensionality of the full fine-tuning gradient is key to closing the performance gap—is both novel and persuasive. The analysis in Section 5.1 and Appendix E effectively characterizes this GID, providing strong empirical evidence for the problem statement.

Strong Empirical Results: The experimental section is a major strength. The authors comprehensively evaluate their methods on a diverse set of tasks (NLU, NLG, Vision) and against a wide array of strong, recent baselines. The results consistently show that RaLoRA and RaLoRA-Pro outperform other LoRA variants and significantly narrow the gap with full fine-tuning, sometimes even surpassing it. The ablation studies (Section 5.2, 5.3, 5.4) are particularly valuable for disentangling the contributions of the different components.

Theoretical Grounding: The paper provides a solid theoretical foundation. The derivation of LoRA as an implicit gradient compressor (Appendix A) is elegant and directly motivates the GID concept. The analysis of the approximation error for RaLoRA's block-diagonal structure (Appendix C) offers a principled explanation for why its architectural change can be more expressive than standard LoRA under a fixed parameter budget.

**Weaknesses:**

**1. Computational Overhead of GID Estimation:** A significant practical concern is the computational cost of estimating GID. The method requires computing the SVD of the full gradient matrix for each target layer over several mini-batches (Algorithm 1). For very large models (e.g., LLaMA 70B), the cost of these SVD operations, even if done only once during a warm-up phase, could be non-trivial. The paper would be strengthened by a discussion or analysis of this overhead, perhaps quantifying the extra time/memory cost compared to standard LoRA initialization.

**2. Clarity on the "One-Shot" vs. "Dynamic" Nature of GID:** The methodology section and Algorithm 1 suggest that the GID estimation and parameter reallocation are performed as a one-time initialization step using gradients accumulated from a few mini-batches. However, the discussion of GID dynamics in Section 5.1 shows that GID evolves during training. This raises questions:

   - Why is a one-time estimation sufficient? Could performance be further improved by periodically re-estimating GID and adapting the structure during training?

   - How sensitive are the results to the specific mini-batches used for the initial GID estimation?
A discussion of these points and perhaps an experiment on the stability of the initial GID estimate would alleviate these concerns.

**3. Hyperparameter Sensitivity and Tuning Cost:** While the proposed methods are shown to be effective, they introduce new hyperparameters: the maximum expansion factor n_max and the rank reallocation range (r_min, r_max). The results in Tables 5 and 6 show that performance is sensitive to these choices, and the authors mention using a grid search. This somewhat offsets the "parameter-efficient" and "easy-to-use" appeal of standard LoRA. A more in-depth discussion on guidelines for setting these hyperparameters or strategies to reduce the tuning cost would be very helpful for practitioners.

**4. Comparison to Other High-Rank PEFT Methods:** The baselines are comprehensive within the LoRA family. However, it would be valuable to see a comparison with other PEFT methods that also aim for higher rank/more expressive updates but through different mechanisms, such as (IA)³ or perhaps a memory-efficient implementation of a method like DiffPruning. This would better situate the contribution within the broader PEFT landscape.

**Questions:**

1. What is the computational overhead (in time and memory) of the GID estimation phase compared to the standard LoRA initialization? Is this cost negligible for very large models (e.g., >100B parameters)?

2. The GID is estimated once at initialization. Given that Figure 3(c) shows GID evolves, have you experimented with dynamic rank adaptation during training? What were the results and challenges?

3. It is better if the authors can discuss some recent works:

   - RandLoRA: Full-rank parameter-efficient fine-tuning of large models
   - Mlae: Masked lora experts for parameter-efficient fine-tuning
   - Latent Space Factorization in LoRA

4. How does the performance and efficiency of your method compare to simply using a standard LoRA with a much higher rank (e.g., r=64 or 128) to cover the high GID, even if it means a higher parameter count? The results in Appendix E.3 (Table 8) are a start, but a more systematic comparison on a equal-parameter or equal-performance basis would be insightful.

---

> ### Author Response · Authors · 2025-11-21
>
> **W1: Computational Overhead of GID Estimation: A significant practical concern is the computational cost of estimating GID. The method requires computing the SVD of the full gradient matrix for each target layer over several mini-batches (Algorithm 1). For very large models (e.g., LLaMA 70B), the cost of these SVD operations, even if done only once during a warm-up phase, could be non-trivial. The paper would be strengthened by a discussion or analysis of this overhead, perhaps quantifying the extra time/memory cost compared to standard LoRA initialization.**
>
> **Q1: What is the computational overhead (in time and memory) of the GID estimation phase compared to the standard LoRA initialization? Is this cost negligible for very large models (e.g., >100B parameters)?**
>
> Response to W1 and Q1:
>
> We appreciate the reviewer raising this practical concern. We have conducted additional profiling to quantify this overhead and compare it against standard LoRA initialization. While standard LoRA initialization is indeed minimal (requiring approximately **5.21 seconds** and **12 MB** of additional memory for an **8B model**), our analysis demonstrates that **the overhead introduced by GID estimation is modest and remains computationally efficient, even for 70B-scale models**. The GID estimation process consists of **two distinct phases**. We detail the costs for each below:
>
> **1. Full Gradient Matrix Computation**
> To minimize memory footprint, we adopt an **optimizer-free strategy** (without storage of optimizer states). We compute the full gradient matrix layer-by-layer, accumulate the gradients over the calibration batch, and offload them to CPU memory on the fly. This ensures minimal peak GPU memory usage. With 1024 samples for estimating GID, the computational cost of our method is shown below:
>
> - **For 8B Models:** The computational cost for this phase is **26.67 seconds**, with a peak GPU memory occupation of **27,355 MB**.
> - **For 70B Models:** We integrate our method with **ZeRO-3 optimization**, which shards the model and gradients across GPUs. In an experiment using 8 GPUs, the computational cost is **57.35 seconds**, with a peak GPU memory usage of **36,139 MB**. This demonstrates that our approach scales **effectively to large models within standard hardware constraints**.
>
> **2. GID Estimation (SVD Calculation)**
> The primary concern lies in the SVD computation. However, this is a **one-time cost** incurred only before training begins. Furthermore, for very large matrices, we employ efficient approximation algorithms.
>
> - **For 8B Models:** The SVD operation on the gradient matrices takes 1 **minute 18.83** **seconds**.
> - **For 70B Models:** The SVD operation on the gradient matrices takes **3 minutes 21.72** **seconds**.
> - **Scalability to >100B Parameters:** Due to the time limit and computational cost, we were unable to conduct experiments on models exceeding 100B parameters. However, the scalability demonstrated by our 70B results provides strong evidence that our findings are applicable to models of 100B scale and beyond.
>
> **3. Conclusion**
> While GID estimation introduces a setup cost compared to the near-instantaneous initialization of vanilla LoRA, the total time (**$\approx$ 100 seconds for 8B and $\approx$ 260 seconds for 70B**) is **negligible** **compared to the total training duration**, which typically spans hours or days. **Given the significant performance gains and improved parameter efficiency demonstrated in our paper (Table 1-3), we believe this one-time initialization overhead is a highly favorable trade-off**.

---

> ### Author Response · Authors · 2025-11-21
>
> **W2: Clarity on the "One-Shot" vs. "Dynamic" Nature of GID: The methodology section and Algorithm 1 suggest that the GID estimation and parameter reallocation are performed as a one-time initialization step using gradients accumulated from a few mini-batches. However, the discussion of GID dynamics in Section 5.1 shows that GID evolves during training. This raises questions:
> Why is a one-time estimation sufficient? Could performance be further improved by periodically re-estimating GID and adapting the structure during training?
> How sensitive are the results to the specific mini-batches used for the initial GID estimation? A discussion of these points and perhaps an experiment on the stability of the initial GID estimate would alleviate these concerns.**
>
> Response to W2:
>
> We appreciate the reviewer’s insightful questions and the opportunity to clarify our approach.
>
> **1. A 2.1**: **We provide a detailed answer to this in Q2**, and summarize below.
>
> We chose to focus on a static pre-assignment for the following reasons: **computational efficiency, stability of GID, and training stability**.
>
> **2. A 2.2**: We computed the estimated GID and subsequent fine-tuning performance across different mini-batch sampling strategies and three distinct random seeds. The results are summarized in the following table.
>
> | mini-batch sampling | seed 1 | seed 2 | seed 3 | GSM8K |
> | --- | --- | --- | --- | --- |
> | sampling 1 | 184.92 | 184.92 | 184.91 | 72.24 $\pm$ 0.29 |
> | sampling 2 | 186.36 | 186.35 | 186.38 | 72.31 $\pm$ 0.44 |
> | sampling 3 | 185.26 | 185.27 | 185.24 | 72.28 $\pm$ 0.24 |
>
> **Key Observations:**
>
> • **Stability:** The estimator proves **highly robust**. The maximum fluctuation in GID across different seeds and sampling batches is **negligible** ($\Delta \approx$  1.46) relative to the magnitude of the dimension ($\approx$ 185).
>
> • **Consistent Structural Gap:** The estimated GID consistently **remains significantly higher than standard fixed LoRA ranks (typically $r=8$)**, reinforcing our paper's core premise that a fixed low rank is insufficient.
>
> • **Robust Downstream Performance:** Most importantly, the downstream performance on GSM8K remains stable and superior (**$\sim$72.2%**) across all configurations, **consistently outperforming the vanilla LoRA baseline (67.78%)**.

---

> ### Author Response · Authors · 2025-11-21
>
> **W3: Hyperparameter Sensitivity and Tuning Cost: While the proposed methods are shown to be effective, they introduce new hyperparameters: the maximum expansion factor n_max and the rank reallocation range (r_min, r_max). The results in Tables 5 and 6 show that performance is sensitive to these choices, and the authors mention using a grid search. This somewhat offsets the "parameter-efficient" and "easy-to-use" appeal of standard LoRA. A more in-depth discussion on guidelines for setting these hyperparameters or strategies to reduce the tuning cost would be very helpful for practitioners.**
>
> Response to W3:
>
> We thank the reviewer for this constructive suggestion. We agree that discussing guidelines for hyperparameter settings is crucial for reducing tuning costs and maintaining the "easy-to-use" appeal of our method.
>
> We presented the sensitivity analysis in Tables 5 and 6. In the revised manuscript, **we will strengthen this discussion and provide detailed strategies to guide practitioners and reduce tuning efforts**, as summarized below.
>
> First, regarding the **maximum expansion factor** $n_{max}$, **we observe that the optimal setting is proportional to the Mean Gradient Intrinsic Dimension (mGID) of the dataset**. A simple and effective guideline is to set $n_{max}$ such that the product of the base rank and the expansion factor, **$r \times n_{max}$, approximates the mGID** of the task.
> As shown in the table below (where $r=8$):
>
> - For tasks with lower mGID (e.g., GSM8K, mGID $\approx$ 178), $8 \times 16 = 128$ is closest to 178, and indeed $n_{max}=16$ yields the best performance.
> - For tasks with higher mGID (e.g., WizardLM, mGID  $\approx$  404), $8 \times 64 = 512$ is closest to 404, making $n_{max}=64$ the optimal choice.
> This principle significantly narrows the search space for $n_{max}$.
>
> | Method | GSM8K (mGID $\approx$ 178) | HuamEval (mGID $\approx$ 269) | WizardLM (mGID $\approx$ 404) |
> | --- | --- | --- | --- |
> | LoRA(r=8) | 67.78 $\pm$ 1.25 | 43.09 $\pm$ 0.35 | 6.15 $\pm$ 0.02 |
> | RaLoRA($n_{max}$=16) | **72.25 $\pm$ 0.53** | 45.12 $\pm$ 1.22 | 6.18 $\pm$ 0.12 |
> | RaLoRA($n_{max}$=32) | 71.42 $\pm$ 0.87 | 47.76 $\pm$ 1.27 | 6.19 $\pm$ 0.2 |
> | RaLoRA( $n_{max}$ =64) | 71.19 $\pm$ 0.79 | **48.78 $\pm$ 1.61** | **6.38 $\pm$ 0.07** |
>
> Second, regarding the **rank reallocation range** $(r_{min}, r_{max})$, we designed three candidate ranges: $(r/2, 2r)$, $(3r/4, 4r)$, and $(r/2, 4r)$. The results indicate that the choice of range **follows a consistent pattern with task complexity (mGID)**:
>
> - **Lower Complexity** (e.g., GSM8K): A relatively narrow range is preferred, specifically $(r/2, 2r)$.
> - **Medium Complexity** (e.g., HumanEval): A medium allocation range is optimal, specifically $(3r/4, 4r)$.
> - **High Complexity** (e.g., WizardLM): A broader range is necessary to capture complex features, specifically $(r/2, 4r)$.
>
> |  | GSM8K (mGID$\approx$178) | HuamEval (mGID $\approx$ 269) | WizardLM (mGID $\approx$ 404) |
> | --- | --- | --- | --- |
> | LoRA(r=8) | 67.78 $\pm$ 1.25 | 43.09 $\pm$ 0.35 | 6.15 $\pm$ 0.02 |
> | RaLoRA-Pro(8-8) | 71.42 $\pm$ 0.87 | 47.76 $\pm$ 1.27 | 6.19 $\pm$ 0.2 |
> | RaLoRA-Pro(4-16) | **72.23 $\pm$ 0.32** | 46.95 $\pm$ 1.06 | 6.12 $\pm$ 0.08 |
> | RaLoRA-Pro(4-32) | 71.39 $\pm$ 1.02 | 45.32 $\pm$ 1.76 | **6.70 $\pm$ 0.23** |
> | RaLoRA-Pro(6-32) | 71.37 $\pm$ 0.88 | **48.17 $\pm$ 0.61** | 6.65 $\pm$ 0.30 |
>
> In summary, **while our proposed methods introduce new hyperparameters, their optimal settings exhibit a strong positive correlation with task complexity**. By following these straightforward guidelines based on mGID, practitioners can effectively reduce tuning costs while leveraging the enhanced performance of our parameter-efficient approach.

---

> ### Author Response · Authors · 2025-11-21
>
> **W4: Comparison to Other High-Rank PEFT Methods: The baselines are comprehensive within the LoRA family. However, it would be valuable to see a comparison with other PEFT methods that also aim for higher rank/more expressive updates but through different mechanisms, such as (IA)³ or perhaps a memory-efficient implementation of a method like DiffPruning. This would better situate the contribution within the broader PEFT landscape.**
>
> Response to W4:
>
> We thank the reviewer for this insightful suggestion. To better situate our contributions within the broader PEFT landscape, we have conducted additional experiments with two representative non-LoRA PEFT methods that employ distinct adaptation mechanisms:
>
> **$\text{(IA)}^3$** [1]: This method adapts the model by learning vectors that rescale the activations at specific layers (Key, Value, and FFN), enabling efficient adaptation with very few parameters.
>
> **FourierFT** [2]: This approach learns a sparse update in the spectral domain via the Discrete Fourier Transform, thereby enabling efficient fine-tuning of large models with fewer parameters.
> The results on GSM8K and HumanEval are presented below:
>
> | Methods | GSM8K | HumanEval |
> | --- | --- | --- |
> | $\text{(IA)}^3$ | 66.62 $\pm$ 1.01 | 0 |
> | FourierFT | 52.71 $\pm$ 4.69 | 0 |
> | LoRA | 67.78 $\pm$ 1.25 | 43.09 $\pm$ 0.35 |
> | RaLoRA (Ours) | 72.25 $\pm$ 0.59 | **48.78 $\pm$ 1.61** |
> | RaLoRA-Pro (Ours) | **73.01 $\pm$ 0.53** | 48.37 $\pm$ 2.54 |
>
> **Analysis:**
>
> From the table, we observe that LoRA methods, especially RaLoRA and RaLoRA-Pro, consistently perform better than (IA)³ and FourierFT, **highlighting the contribution of our work in narrowing the gap between PEFT and full fine-tuning**.
> Both $\text{(IA)}^3$ and **FourierFT** did not converge on the **Code-Feedback** dataset. We believe this may be due to the complexity of the task, as indicated by the **higher GID** in our paper (Figure 3b and Table 5).
>
> **Experimental Settings:**
>
> We utilized the official implementations from the PEFT library with hyperparameters reported in their respective papers:
>
> • **$\text{(IA)}^3$:** lr=3e-3, adapting modules ["q", "k", "v", "o", "W3"].
>
> • **FourierFT:** lr=3e-2, n_frequency=1000, scaling=300, adapting modules ["q", "k", "v", "o"].
>
> We will include these additional comparisons and discussions in the final version of the paper.
>
> [1] Liu, Haokun, et al. "Few-shot parameter-efficient fine-tuning is better and cheaper than in-context learning." NeurIPS (2022).
>
> [2] Gao, Ziqi, et al. "Parameter-efficient fine-tuning with discrete fourier transform." ICML (2024).
>
> **Q2: The GID is estimated once at initialization. Given that Figure 3(c) shows GID evolves, have you experimented with dynamic rank adaptation during training? What were the results and challenges?**
>
> Response to Q2:
>
> We appreciate the reviewer’s question. We have not tested dynamic rank scheduling during training, as we chose to focus on static pre-assignment for the following reasons:
>
> 1. **Computational Efficiency and Stability of GID**: As shown in **Figure 3(c)** of our paper, **the GID remains relatively stable during training**. Given this stability, introducing dynamic rank adjustments would require recalculating the GID multiple times throughout the training process, **which would add unnecessary computational overhead without yielding significant performance gains**. Therefore, we chose to use a static pre-assignment approach, which ensures both computational efficiency and simplicity.
> 2. **Training Stability and Practicality**: **Dynamically** **altering the LoRA matrix structure during training could lead to instability**, potentially undermining the practical usability of the method. For example, AdaLoRA [1], a popular method that dynamically adjusts ranks during training, uses a **masking technique** **instead of directly modifying the LoRA matrix structure**. This approach suggests that dynamically changing the matrix rank could introduce challenges, particularly regarding stability and generalization. As a result, we chose to maintain a static rank assignment to avoid these potential pitfalls.
>
> [1] Zhang, Qingru, et al. "Adalora: Adaptive budget allocation for parameter-efficient fine-tuning." *arXiv preprint arXiv:2303.10512* (2023).

---

> ### Author Response · Authors · 2025-11-21
>
> **Q3: It is better if the authors can discuss some recent works:**
>
> **- RandLoRA: Full-rank parameter-efficient fine-tuning of large models**
>
> **- Mlae: Masked lora experts for parameter-efficient fine-tuning**
>
> **- Latent Space Factorization in LoRA**
>
> Response to Q3:
>
> Thank you for the constructive feedback. We will elaborate on these works in detail in the revised manuscript and summarize our thoughts as follows:
>
> The paper "Latent Space Factorization in LoRA" is referred to as the "FVAE-LoRA" method proposed in this paper.
>
> - **RandLoRA**'s core motivation is to perform full-rank updates by introducing a linear combination of low-rank, non-trainable random matrices. This approach still belongs to the **rank augmentation** category discussed in our paper. While this method enhances rank, **it does so heuristically** and neglects the intrinsic dimensionality difference between LoRA and the FFT gradient, which may lead to suboptimal utilization of the parameters.
>
> - **MLAE** and **FVAE-LoRA** are both methods designed to address redundancy in the LoRA. **MLAE** introduces a masking strategy to enhance the independence and diversity of the low-rank matrices, while **FVAE-LoRA** incorporates a variational autoencoder (VAE) framework to explicitly separate task-relevant information from residual information. However, both methods focus primarily on removing redundancy within the low-rank update subspace of LoRA. While removing redundancy in low-rank matrices can improve performance, it does not fully unlock the model's potential, leading to inefficient use of computational resources.
>
> - **Our Distinction:** Our work is **the first to explicitly address the** **mismatch between LoRA and FFT in terms of GID**. We introduce **a novel entropy-based estimator to accurately measure GID**, and propose an **adaptive block-diagonal structure** to align LoRA’s representational capacity with the estimated GID across layers, while keeping parameter count fixed. Furthermore, we introduce **RaLoRA-Pro**, which **jointly performs intra-layer GID alignment and inter-layer importance-aware rank allocation**. These **adaptive designs** enable more effective alignment with FFT and lead to further performance improvements.
>
> **Q4: How does the performance and efficiency of your method compare to simply using a standard LoRA with a much higher rank (e.g., r=64 or 128) to cover the high GID, even if it means a higher parameter count? The results in Appendix E.3 (Table 8) are a start, but a more systematic comparison on a equal-parameter or equal-performance basis would be insightful.**
>
> Response to Q4:
>
> We appreciate the reviewer’s insightful suggestion. We have expanded the results in Appendix E.3 (Table 8). In LoRA+GID, we set the LoRA rank for each layer to the GID. The experimental range covers a broad spectrum of parameter counts, **from smaller to larger than that of LoRA+GID**, as shown in the following table.
>
> **Key Findings:**
>
> - **LoRA+GID** significantly outperforms LoRA with different ranks, effectively capturing the heterogeneity in update directions between layers, thus offering better alignment capabilities.
> - **When LoRA's rank is smaller than the mGID, increasing the rank results in significant performance improvements**. However, **when the rank exceeds mGID, performance levels off or even slightly declines** (e.g., on GSM8K), suggesting that using a rank smaller than mGID may lose important information, while a rank greater than mGID may introduce redundant noise.
>
> |  | GSM8K |  |  | HummanEval |  |
> | --- | --- | --- | --- | --- | --- |
> | Method | Params | Acc | Method | Params | Acc |
> | FFT | 1.34B | 73.69 $\pm$ 0.28 | FFT | 1.34B | 51.63 $\pm$ 1.27 |
> | LoRA(r=8) | 6.82M | 67.78 $\pm$ 1.25 | LoRA(r=8) | 6.85M | 43.09 $\pm$ 0.35 |
> | LoRA(r=128) | 109.05M | 72.88 $\pm$ 0.32 | LoRA(r=128) | 109.05M | 45.73 $\pm$ 1.22 |
> | LoRA(r=256) | 218.10M | 73.49 $\pm$ 0.42 | LoRA(r=256) | 218.10M | 48.17 $\pm$ 2.66 |
> | LoRA(r=200) | 170M | 74.12 $\pm$ 1.23 | LoRA(r=380) | 323M | 49.39 $\pm$ 2.66 |
> | LoRA(r=400) | 340.79M | 73.99 $\pm$ 0.33 | LoRA(r=400) | 340.79M | 49.59$\pm$1.27 |
> | LoRA+GID | 175M | **75.41 $\pm$ 0.84** | LoRA+GID | 323M | **52.64 $\pm$ 1.27** |

---

> ### Comment · Reviewer_4tTg · 2025-11-22
>
> The authors have addressed my concerns. I have therefore revised my score accordingly.

---

> ### Author Response · Authors · 2025-11-22
>
> Thank you very much for your thoughtful follow-up and for considering our response. We sincerely appreciate your time, constructive feedback, and your decision to raise the score to 8.

---

### Official Review · Reviewer_rn9v · 2025-10-31

**Soundness:** 3
**Presentation:** 3
**Contribution:** 2
**Rating:** 6
**Confidence:** 4

**Summary:**

This paper proposes RaLoRA and RaLoRA-Pro, methods for adapting the rank of LoRA adapters by estimating a new quantity called Gradient Intrinsic Dimensionality (GID) using an entropy-based “effective rank” estimator. The idea is that LoRA’s fixed low-rank subspace can misalign with the true gradient subspace of full fine-tuning; thus, estimating GID per layer and adjusting ranks could better use the parameter budget. The paper reports strong improvements on GLUE, MT-Bench, GSM8K, HumanEval, and image classification.

**Strengths:**

1. The problem motivation, mismatch between LoRA rank and true gradient structure, is interesting.
The core insight of this block-diagonal design hinges on the property that $\text{Rank}(BA)= \sum_{i=1}^{n_l} rank(B_iA_i)$. By employing block diagonalization, the authors aim to strategically augment the overall $\text{Rank}(BA)$ to align with the estimated Gradient Intrinsic Dimension (GID).

2. The proposed entropy-based GID estimator is conceptually clean and grounded in established ideas.

3. Writing is mostly clear and well-organized.

**Weaknesses:**

1. The paper estimates Gradient Intrinsic Dimensionality using spectral entropy, which typically requires SVD and significant computation. Is SVD performed for every layer at each step, or precomputed and frozen? What is the time complexity compared to other estimation methods? It would be helpful to show robustness under different batches to assess the stability of the gradient SVD.

2. The design of the block diagonal matrix may restrict gradient flow and information sharing.  If the goal is merely to increase $\text{Rank}(BA)$, other approaches such as introducing a Hadamard product (as in HiRA [1], Eq. (4)) could also achieve a higher rank. A comparative discussion would further elucidate the distinctions between these approaches.

3. The visualization of importance estimation in RaLoRA-Pro is unclear. How does it differ from prior works that allocate LoRA ranks based on importance or sensitivity?

4. The paper adopts a dynamic rank allocation strategy. Has it been fairly compared with other methods, for example, in terms of parameter count or efficiency?

5. Typos:
     1) Chapter “Disscusion” → “Discussion”
     2) Inconsistent capitalization of “GSM8K” (e.g., “GSM8k” in Table 2). Please standardize.
     3) All equation references should use “Equation XX” instead of “equation XX”.

[1] HiRA: Parameter-Efficient Hadamard High-Rank Adaptation for Large Language Models

**Questions:**

Q1: The paper estimates Gradient Intrinsic Dimensionality (GID) using spectral entropy, which requires SVD decomposition.
Is SVD computed for every layer at each update step, or is it precomputed and then frozen?
What is the time complexity compared with other baseline methods?
Q2: The current block-diagonal design may restrict gradient flow and information sharing. Could the authors discuss or compare this design with HiRA, or provide further justification for choosing this particular formulation?
Q3: The visualization of importance estimation in RaLoRA-Pro is unclear. How does it differ from prior works that allocate LoRA ranks based on importance or sensitivity?
Q4: The paper proposes a dynamic rank allocation method. Has it been fairly compared with other rank adaptation approaches? Could the authors analyze the comparison in terms of parameter count or efficiency?

---

> ### Author Response · Authors · 2025-11-21
>
> **W1 & Q1: The paper estimates Gradient Intrinsic Dimensionality using spectral entropy, which typically requires SVD and significant computation. Is SVD performed for every layer at each step, or precomputed and frozen? What is the time complexity compared to other estimation methods? It would be helpful to show robustness under different batches to assess the stability of the gradient SVD.**
>
> Response to W1 and Q1:
>
> We appreciate the reviewer’s question.  To clarify, the SVD required for estimating Gradient Intrinsic Dimensionality (GID) is computed **once** before training begins and **remains fixed** throughout the training process.
>
> **Time Complexity of SVD Computation:**
> The time complexity of the SVD calculation is $O(min⁡(din,dout)^3)$, which occurs only **once at the** **precomputation phase** before training starts. This approach avoids the computational overhead that would result from recalculating GID at each update step.
>
> **Time Efficiency Compared with baseline methods:**
> We also conducted extensive benchmarking to compare the time efficiency of our method with other baseline methods in the following table. The **key difference** is that our proposed method includes an additional **GID estimation phase**. As a result, we report the time spent on GID estimation, in addition to the training time.
>
> - **GID Estimation Phase**: This phase takes approximately **1 minute 45.5 seconds**, with **1 minute 18.83 seconds** spent on computing the SVD.
> - **Training Time**: Despite the additional GID estimation phase, **our methods (32 minutes 29.36 seconds) are as efficient as the baselines ($\approx$30 minutes)**, with only a **~3-minute difference** compared to the fastest baseline, MELoRA, and even faster than MoRA (**37 minutes 15.80 seconds**).
> - Given the significant performance improvements achieved by RaLoRA and RaLoRA-Pro, they offer an **excellent trade-off between performance and efficiency**.
>
> | Method | efficiency (MetaMathQA) | GSM8K | HumanEval |
> | --- | --- | --- | --- |
> | LoRA | 30 minutes 3.81 seconds | 67.78$\pm$ 1.25 | 43.09$\pm$0.35 |
> | MELoRA | 29 minutes 11.49 seconds | 71.27$\pm$1.1 | 45.73$\pm$2.80 |
> | MoRA | 37 minutes 15.80 seconds | 68.12$\pm$0.91 | 29.68$\pm$5.73 |
> | AdaLoRA | 30 minutes 26.25 seconds | 70.63$\pm$0.77  | 41.46$\pm$3.66 |
> | RaLoRA (Ours) | 32 minutes 29.36 seconds | 72.25$\pm$0.59 | 48.78$\pm$1.61 |
> | RaLoRA-Pro (Ours) | 30minutes 44.63 seconds | 73.01$\pm$0.53 | 48.37 $\pm$2.54 |

---

> ### Author Response · Authors · 2025-11-21
>
> **W2 & Q2: The current block-diagonal design may restrict gradient flow and information sharing. Could the authors discuss or compare this design with HiRA, or provide further justification for choosing this particular formulation?**
>
> Response to W2 and Q2:
>
> We would like to thank the reviewer for the valuable feedback. We have compared our approach with HiRA from both theoretical and experimental perspectives. We will incorporate this discussion into the final version of the paper.
>
> 1. **Theoretical Foundation：**
> - HiRA is a **rank augmentation-based method** that we mentioned in our paper, which aims to **address the low-rank limitations** in complex tasks via the **Hadamard product**. Many experiments have demonstrated that HiRA can effectively increase the rank of update parameter matrices without introducing extra trainable parameters. However, its approach to rank enhancement is heuristic, and it ignores the fundamental intrinsic dimensional differences and the gap between LoRA and FFT.
> - In contrast, the method proposed in this paper represents a **novel approach distinct from** *rank augmentation*. The goal of our method is to **bridge the performance gap with FFT by aligning the LoRA rank to the layer-wise estimated GID**. The theory is grounded in the idea that LoRA acts as a gradient compressor (see Appendix A). **When the LoRA rank is significantly lower than the GID, substantial information is lost**. To address this, **we propose an efficient entropy-based GID estimator**, which captures the structural information of the gradient during fine-tuning for a specific task. The GID estimator is **conceptually orthogonal to most existing LoRA variants and can even benefit a broad range of PEFT methods**.
>
> The current block-diagonal design of RaLoRA was chosen to align with the estimated GID without increasing the additional trainable parameters. In Appendix E.3 of the paper, we present experiments with **LoRA+GID**, showing that it significantly outperforms LoRA with comparable parameters on tasks like GSM8K and HumanEval, **further demonstrating the effectiveness and orthogonality of GID**.
>
> 2. **Experimental comparison:**
>
> We experiment to compare HiRA with our methods for different ranks. To ensure a fair comparison, we adopt the learning rate (1e-3) configuration reported in the HiRA paper.  The experimental results shown below demonstrate that **our methods outperform HiRA in both GSM8k and HumanEval**.
>
> | Method | GSM8K | HumanEval |
> | --- | --- | --- |
> | LoRA(r=8) | 67.78$\pm$ 1.25 | 43.09$\pm$0.35 |
> | HiRA(r=8) | 69.93 $\pm$ 0.39 | 45.73 $\pm$ 1.22 |
> | RaLoRA(r=8) | 72.25 $\pm$ 0.59 | 48.78 $\pm$ 1.61 |
> | RaLoRA-Pro(r=8) | 73.01 $\pm$ 0.53 | 48.37 $\pm$ 2.54 |
>
> We will incorporate this discussion into the final version of the paper.

---

> ### Author Response · Authors · 2025-11-21
>
> **W3 & Q3: The visualization of importance estimation in RaLoRA-Pro is unclear. How does it differ from prior works that allocate LoRA ranks based on importance or sensitivity?**
>
> Response to W3 and Q3:
>
> Thank you for raising this important question. We apologize for not presenting the visualization of importance estimation in RaLoRA-Pro more clearly. In the revised version, **we will strengthen the description and explanation of this process, particularly in Section 3.4 and Figure 2**, to ensure that it is presented more clearly.
>
> **Regarding the difference between our method and prior rank allocation works**:
> RaLoRA-Pro **is not merely a reallocation of parameters;** it represents a fundamental shift from simple **"Resource Allocation"** to **"FFT Alignment."**
>
> The **distinction** can be understood through the following progressive levels of our contribution:
>
> **1. Foundation: Identifying the Structural mismatch via GID (Gradient Intrinsic Dimensionality)**
> Prior rank allocation methods typically rely solely on "importance" (i.e., sensitivity, how much a layer contributes to the loss). While useful, they overlook a critical structural issue: **the mismatch between LoRA’s low-rank constraint and the high-dimensional nature of Full Fine-Tuning (FFT) gradients (see proof in Appendix A).**
>
> - **Our Discovery:** As shown in **Figure 1(a)** and **Figure 3**, the intrinsic dimensionality of FFT gradients often reaches up to 300, whereas standard LoRA is restricted to a rank of roughly 8.
> - **Our Innovation:** We introduce a novel **entropy-based GID estimator**. Unlike importance scores, this estimator quantifies he **true effective update directions required by each layer**. This estimator is **orthogonal** to importance scores and existing methods (see Appendix E.3).
>
> **2. Mechanism: RaLoRA as a General Extension of LoRA**
>
> - **The Generalization:** RaLoRA acts as a **general extension of LoRA**. By employing a block-diagonal decomposition guided by our GID estimator, **RaLoRA adaptively adjusts the number of blocks $n_l$  per layer**:
>
>     - **Low GID ($n_l \to 1$):** RaLoRA reduces mathematically to standard **Vanilla LoRA**, focusing on dominant directions.
>     - **High GID ($n_l > 1$):** RaLoRA expands the **effective rank** (e.g., from $r \to n_l \times r$) without increasing the parameter budget.
>
> - **The Benefit:** This design allows RaLoRA to **adaptively trade fidelity to dominant directions for broader expressiveness across heterogeneous subspaces** (see proof in **Appendix C**).
>
> **3. The Comprehensive Framework: RaLoRA-Pro (Dual Adaptive Alignment)**
>
> This leads to the unique contribution of RaLoRA-Pro, **which is the first framework to unify inter-layer importance with intra-layer geometric alignment**.
>
> - **Prior Work (Single Alignment):** Methods like AdaLoRA reallocate parameter budgets from unimportant layers to important ones. **However, even if a layer is assigned a higher rank (e.g., $r=64$) via reallocation, it may still fail to capture a gradient subspace with a GID of 300 efficiently.**
>
> - **RaLoRA-Pro (Dual Alignment):** We **optimize two distinct dimensions simultaneously to maximize parameter efficiency**:
>
>     - **Inter-layer (Global Importance):** We use loss sensitivity to determine the **parameter budget** $r_{ref}$ for each layer, ensuring influential layers receive more capacity.
>
>     - **Intra-layer (Local Geometry):** Crucially, we then **use the GID estimator to align with the FFT gradient structure** $n_l$.
>
> In summary, **RaLoRA-Pro differs from prior works because it is a dual alignment of FFT intra and inter layers**. This dual strategy allows us to bridge the performance gap between LoRA and FFT more effectively, as evidenced by the superior performance (Tables 1-3).

---

> ### Author Response · Authors · 2025-11-21
>
> **W4 & Q4: The paper proposes a dynamic rank allocation method. Has it been fairly compared with other rank adaptation approaches? Could the authors analyze the comparison in terms of parameter count or efficiency?**
>
> Response to W4 and Q4:
>
> We thank the reviewer for this insightful question. We ensured a strictly fair comparison across all experiments. As detailed in our experimental setup (Section 4 and Appendix G) and summarized below:
>
> - **Hyperparameter configurations**: The hyperparameter configurations for all baselines in our paper (including MELoRA, MoRA, and AdaLoRA) were **from established baselines or standardized following their original papers**.
> - **Parameter count**: The total trainable parameter budget for every method was controlled to be **comparable to Vanilla LoRA (rank $r=8$) as far as possible**.
>
> As shown in the following table, **our methods maintain a training efficiency comparable to all baselines (only ~3 minutes difference compared to the fastest baseline, MELoRA, and faster than MoRA)** **while** **maintaining or reducing the parameter count**. Given the significant performance gains reported in the paper, **RaLoRA and RaLoRA-Pro** **offer a superior trade-off between performance and efficiency**.
>
> | Method | Params (MetaMathQA) | efficiency (MetaMathQA) | GSM8K | HumanEval |
> | --- | --- | --- | --- | --- |
> | LoRA | 6.82M | 30 minutes 3.81 seconds | 67.78 $\pm$ 1.25 | 43.09 $\pm$ 0.35 |
> | MELoRA | 6.82M | 29 minutes 11.49 seconds | 71.27 $\pm$ 1.1 | 45.73 $\pm$ 2.80 |
> | MoRA | 6.82M | 37 minutes 15.80 seconds | 68.12 $\pm$ 0.91 | 29.68 $\pm$ 5.73 |
> | AdaLoRA | 10.23M(init) drop to $\approx$ 6.82M(final) | 30 minutes 26.25 seconds | 70.63 $\pm$ 0.77  | 41.46 $\pm$ 3.66 |
> | RaLoRA (Ours) | 6.82M | 32 minutes 29.36 seconds | 72.25 $\pm$ 0.59 | 48.78 $\pm$ 1.61 |
> | RaLoRA-Pro (Ours) | 6.38M | 30minutes 44.63 seconds | 73.01 $\pm$ 0.53 | 48.37 $\pm$ 2.54 |
>
> **W5: Typos**
>
> Response to W5:
>
> We sincerely thank the reviewer for their meticulous review of our work. We apologize for the typographical and formatting issues pointed out in the review. We have corrected these errors in the manuscript. We have also carefully rechecked the entire manuscript to ensure similar errors do not persist.

---

### Official Review · Reviewer_keQx · 2025-10-31

**Soundness:** 3
**Presentation:** 3
**Contribution:** 3
**Rating:** 6
**Confidence:** 3

**Summary:**

This paper introduces **Gradient Intrinsic Dimensionality (GID)** to explain the performance gap between low-rank adaptation (LoRA) and full fine-tuning (FFT). The authors argue that LoRA’s fixed low-rank subspace fails to align with the true gradient subspace of FFT. To address this, they propose **RaLoRA**, which estimates each layer’s effective gradient rank via an entropy-based metric and dynamically allocates LoRA capacity accordingly. A further variant, **RaLoRA-Pro**, performs inter-layer reallocation guided by loss sensitivity. Experiments on diverse NLP, code, and vision tasks show consistent and sometimes significant gains over existing PEFT methods without increasing parameters.

**Strengths:**

1. **Clear and meaningful motivation.**
The paper addresses a known limitation of LoRA—its rigid and sometimes misaligned update subspace—through a theoretically motivated concept (GID). This adds a fresh geometric interpretation to the efficiency–performance trade-off in PEFT.

2. **Methodologically elegant design.**
The entropy-based estimator and adaptive rank allocation are conceptually simple yet principled. The proposed dual-level alignment (intra- and inter-layer) is coherent and implementable without modifying the base model.

3. **Comprehensive experimental evaluation.**
Results span multiple domains and include comparisons to strong baselines such as DoRA, LoRA+, PiSSA, and MELoRA. The reported gains are consistent and statistically significant, and the ablation analyses are generally convincing.  The ablation studies and visualizations (e.g., GID heatmaps) support the empirical narrative.

4. **Readable and reproducible.**
The algorithm is clearly described, and the proposed method could be implemented with minimal modification to existing LoRA frameworks. The method can potentially generalize to other PEFT frameworks and adaptive subspace modeling approaches, making it of interest to the broader community.

**Weaknesses:**

**1. Incremental novelty.**
Although the paper introduces Gradient Intrinsic Dimensionality (GID) and rank alignment, the overall idea aligns closely with recent adaptive-PEFT variants such as GoRA, LoRA+, and DoRA. The contribution mainly reframes known intuitions in a new terminology without offering a fundamentally new optimization mechanism or theoretical insight.

**2. Lack of theoretical rigor.**
While GID is intuitively appealing, the paper lacks formal analysis linking GID alignment to optimization dynamics or generalization improvement. No convergence guarantees, projection error bounds, or theoretical explanations are provided for why matching ranks to GID improves performance.


**3. Computational overhead and scalability.**
The GID estimator requires repeated singular-value decompositions (SVDs) on layerwise gradients, which can become expensive at scale. The paper omits concrete measurements of wall-clock overhead, GPU memory usage, or FLOP ratios compared with vanilla LoRA. It remains unclear whether RaLoRA scales to 70B-parameter LLMs or multi-GPU fine-tuning setups.

**4. Interpretability and analytical depth.**
The visualization of GID heatmaps provides little insight into model behavior. The work does not analyze whether GID correlates with layer function, curvature, or Fisher information. A deeper discussion of why certain layers have high or low GID would strengthen the scientific contribution.

**Questions:**

1. How stable is the entropy-based GID estimator across mini-batch sampling, different random seeds, or noisy gradients?


2. What is the computational overhead (in wall-clock time, GPU memory, or FLOPs) relative to standard LoRA?


3. Does GID correlate with curvature or layer sensitivity measured via Fisher Information or Hessian spectra?


4. Could the observed improvements of RaLoRA be attributed to implicit regularization effects rather than true gradient alignment?


5. Have you tested dynamic rank scheduling during training instead of static pre-assignment?

---

> ### Author Response · Authors · 2025-11-21
>
> **W1: Incremental novelty. Although the paper introduces Gradient Intrinsic Dimensionality (GID) and rank alignment, the overall idea aligns closely with recent adaptive-PEFT variants such as GoRA, LoRA+, and DoRA. The contribution mainly reframes known intuitions in a new terminology without offering a fundamentally new optimization mechanism or theoretical insight.**
>
> Response to W1:
>
> We sincerely thank you for your thoughtful and detailed feedback. We greatly appreciate the time and effort you have put into reviewing our work, and we would like to kindly clarify a few points in response to your observations.
>
> ### Distinction from Existing Adaptive-PEFT Variants
>
> 1. **Comparison with optimization methods**: **LoRA+** (adjusting learning rate ratios) and **DoRA** (decoupling magnitude and direction updates) focus on optimizing training dynamics. However, **even with** **optimal optimization**, these methods are fundamentally limited: **they cannot represent a weight update whose intrinsic rank exceeds $r$**, as shown in Appendix C.2.
> 2. **Comparison with Rank Augmentation methods: GoRA** and **AdaLoRA** adjust rank allocation based on layer importance. However, **it ignores the intra-layer rank alignment in LoRA**, which we argue is a critical factor (as discussed in Appendix A and in our response to Q2). Another method for enhancing LoRA's rank is by stacking multiple LoRA modules (e.g., MELoRA and ReLoRA). This approach, however, **relies on heuristics and is manually set**. While previous work acknowledges the importance of LoRA's rank, they all overlook a crucial question: **How to bridge the gap between LoRA and full fine-tuning?**
>
> ### Our core contributions
>
> 1. **Theoretical Insight: The Quantification of GID Mismatch**
>
>     1\) A key theoretical contribution of this paper is the identification and quantification of the **significant mismatch between the rank of standard LoRA (usually $r=8$) and the GID (up to 300).**
>
>     2\) We introduce a **novel entropy-based GID estimator** **to systematically measure this gap for the first time**. This estimator provides a **principled metric** rather than a heuristic one.
>
>     3\) It is **conceptually orthogonal** to most existing LoRA variants (as shown in Appendix E.3) and can serve as a plug-in tool to benefit a broad range of PEFT methods.
>
>     4\) We provide both **theoretical** (Appendix A) and **empirical evidence** (Figures 1, Tables 1-3) demonstrating that bridging this specific GID gap is crucial for closing the performance gap between LoRA and FFT.
>
> 2. **Novel Optimization Mechanism: RaLoRA as a Generalized Extension of LoRA (see proof in Appendix C)**
>
>     1\) When GID is **low**, $n_l \to 1$, RaLoRA reduces to standard LoRA, focusing on dominant directions.
>
>     2\) When GID is **high**, RaLoRA increases $n_l$, **adaptively sacrificing fidelity to the dominant directions to gain broader expressiveness** by parallelizing learning across heterogeneous subspaces.
>
>     3\) This is a structural optimization mechanism that dynamically alters the topology of the update matrix based on data complexity.
> 3. **RaLoRA-Pro: First Dual Adaptive Alignment Framework**
>
>     **RaLoRA-Pro**, **which is the first framework to unify inter-layer importance with intra-layer  GID alignment**, allowing for **more efficient utilization of trainable parameters** as evidenced by our empirical results (Tables 1-3).
>
> In conclusion, we would like to politely clarify that our main contribution is not just a reframing of existing methods but **the introduction of GID as a key factor for bridging the performance gap between LoRA and FFT**. **The GID estimator can serve as a plug-in tool**, benefiting most LoRA variants and PEFT methods. We present two novel optimization mechanisms: **RaLoRA, which generalizes LoRA**, and **RaLoRA-Pro, the first dual adaptive alignment framework**. Extensive experiments demonstrate the effectiveness of our approaches (Tables 1-3).

---

> > ### Author Response · Authors · 2025-11-21
> >
> > **W2: Lack of theoretical rigor. While GID is intuitively appealing, the paper lacks formal analysis linking GID alignment to optimization dynamics or generalization improvement. No convergence guarantees, projection error bounds, or theoretical explanations are provided for why matching ranks to GID improves performance.**
> >
> > Response to W2:
> >
> > We thank the reviewer for the insightful comment. We have discussed the question of why aligning GID (Gradient Intrinsic Dimensionality) can improve performance by providing a formal theoretical discussion in two parts.
> >
> > 1. **LoRA acts as a Gradient Compressor** (see Appendix A for details).
> > We analyze the optimization dynamics of LoRA and derive the following formula:
> >
> > $$
> > \Delta(BA) \approx -\eta \left( B_t B_t^\top G_t + G_t A_t^\top A_t \right)
> > $$
> >
> > This reveals that LoRA implicitly projects the full gradient $G_t$ onto a low-rank subspace spanned by $B_t B_t^\top$ and $A_t^\top A_t$. **When the LoRA rank is significantly lower than the intrinsic dimensionality of the gradient matrix, substantial information is lost**. This necessitates a principled and adaptive GID alignment framework to balance efficiency and performance.
> >
> > 2. **RaLoRA is a generalized Extension of LoRA** (see Appendix C for details).
> >
> > When the number of parallel blocks $n_l = 1$, **RaLoRA reduces to LoRA**, which provides **fine-grained approximation along dominant gradient directions**. As $n_l$ increases, RaLoRA adaptively **trades precision in dominant directions for broader expressivity**, capturing a more diverse spectrum of gradient subspaces. **The selection of $n_l$ is adaptively driven by our proposed entropy-based estimator**.
> > Specifically, we derive the theoretical minimum approximation error for LoRA:
> >
> > $$
> > E _ {\text{LoRA}} = \min _ {\mathrm{rank}(BA) \le r} \left\\| \Delta W - BA \right\\| _ F^2 = \sum _ {i=r+1}^{\min(m,n)} \sigma_i^2
> > $$
> >
> > This shows that **LoRA captures only the top $r$ singular directions**. When the gradient exhibits slow spectral decay (i.e., high GID), the tail singular values ( $\sigma_{r+1}, \sigma_{r+2}, \dots$ ) remain significant, resulting in substantial approximation loss.
> >
> > The total approximation error for RaLoRA is:
> >
> >  $\begin{aligned} E _{\text{RaLoRA}}=\underbrace{\sum _{r\neq c}\\|\Delta W _{r,c}\\| _{F}^{2}} _{\text{Error from Truncation}}+\underbrace{\sum _{p=1}^{n _{l}}\\|\sum _{i=1}^{\min(m,n)} \sigma_i u _{i,p} v _{p,r}^\top-A _pB _p\\| _{F}^{2}} _{\text{Error from Approximation}} \end{aligned}$
> >
> > **RaLoRA sacrifices fidelity to global top singular directions but gains by parallelizing learning across heterogeneous subspaces**.

---

> ### Author Response · Authors · 2025-11-21
>
> **Q1:How stable is the entropy-based GID estimator across mini-batch sampling, different random seeds, or noisy gradients?**
>
> Response to Q1:
>
> We are grateful to the reviewers for the valuable question.
>
> To rigorously assess the stability of our entropy-based GID estimator, we conducted comprehensive ablation studies on the GSM8K benchmark. We evaluated the estimator's sensitivity to three key factors: **mini-batch sampling**, **random seed**, and **gradient noise**.
>
> **1. Stability Across mini-batch sampling and Random Seeds:**
>
> We computed the estimated GID and subsequent fine-tuning performance across different mini-batch sampling strategies and three distinct random seeds. The results are summarized in the following table.
>
> | mini-batch sampling | random seed 1 | random seed 2 | random seed 3 | GSM8K |
> | --- | --- | --- | --- | --- |
> | sampling 1 | 184.92 | 184.92 | 184.91 | 72.24 $\pm$ 0.29 |
> | sampling 2 | 186.36 | 186.35 | 186.38 | 72.31 $\pm$ 0.44 |
> | sampling 3 | 185.26 | 185.27 | 185.24 | 72.28 $\pm$ 0.24 |
>
> **Key Observations:**
>
> • **Stability:** The estimator proves **highly robust**. The maximum fluctuation in GID across different seeds and sampling batches is **negligible** ($\Delta \approx$ 1.46) relative to the magnitude of the dimension ($\approx$ 185).
>
> • **Consistent Structural Gap:** The estimated GID consistently **remains significantly higher than standard fixed LoRA ranks (typically $r=8$)**, reinforcing our paper's core premise that a fixed low rank is insufficient.
>
> • **Robust Downstream Performance:** Most importantly, the downstream performance on GSM8K remains stable and superior ($\sim$**72.2%**) across all configurations, **consistently outperforming the vanilla LoRA baseline (67.78%)**.
>
> **2. Robustness to Noisy Gradients:**
>
> To evaluate the estimator's resilience to gradient approximation errors or stochastic noise, we injected varying levels of **Gaussian noise** into the gradients before estimation. **Table 2** reports the impact of varying noise rates on the estimated GID and model performance.
>
> | Noise Rate | Estimated GID | GSM8K Accuracy (%) |
> | --- | --- | --- |
> | 0.1% | 184.92 | 72.2 |
> | 1% | 187.95 | 72.25 |
> | 3% | 193.62 | 72.4 |
> | 5% | 198.41 | 72.21 |
>
> **Key Observations:**
>
> • **Theoretical Consistency:** As the noise rate increases from 0.1% to 5%, the estimated GID rises slightly (from $\approx 185$ to $\approx 198$). **This behavior is theoretically expected**: random noise tends to flatten the singular value spectrum, increasing the spectral entropy.
>
> • **Performance Resilience:** Despite the slight inflation in estimated GID due to noise, the downstream accuracy on GSM8K remains remarkably stable (**fluctuating between 72.2% and 72.4%**).
>
> • **Mechanism of Stability:** Crucially, RaLoRA determines the number of blocks via logarithmic binning ($n_l = 2^{\lfloor \log_2 (\text{erank}/r) \rfloor}$, Eq. 4). **This discretization acts as a buffer, meaning slight fluctuations in the raw GID estimate rarely alter the final architectural configuration**, ensuring stable performance even under noisy conditions.

---

> > ### Author Response · Authors · 2025-11-21
> >
> > **W3: Computational overhead and scalability.The GID estimator requires repeated singular-value decompositions (SVDs) on layerwise gradients, which can become expensive at scale. The paper omits concrete measurements of wall-clock overhead, GPU memory usage, or FLOP ratios compared with vanilla LoRA. It remains unclear whether RaLoRA scales to 70B-parameter LLMs or multi-GPU fine-tuning setups.**
> >
> > **Q2: What is the computational overhead (in wall-clock time, GPU memory, or FLOPs) relative to standard LoRA?**
> >
> > Response to W3 and Q2:
> >
> > Thank you for raising this important point for PEFT. Our method performs **one-time SVD** on layerwise gradients computed using a small fraction of training data before the formal training process. Hence, our method remains efficient compared to vanilla LoRA. **Also, our method can be readily extended to multi-GPU training setups and large-scale model setups**.
> >
> > 1. **Wall-clock overhead, GPU memory usage, and FLOPs**
> >
> >     We also perform comprehensive comparisons. Specifically:
> >
> >     - The **GID estimation phase** takes **1 minute 45.5 seconds** (with **1 minute 18.83 seconds** spent on SVD) and consumes a peak GPU memory of **27,355 MB**.
> >     - The **fine-tuning phase of RaLoRA** takes **32 minutes 29.36 seconds** with a peak GPU memory usage of **27,411 MB**, while LoRA takes **30 minutes 3.81 seconds** with a peak GPU memory usage of **26,569 MB**.
> >     - **FLOPs**: RaLoRA **theoretically maintains FLOPs parity with vanilla LoRA**, as they both have the same number of parameters.
> >
> >     These results confirm that our method introduces **minimal computational overhead**, with less than **5\% additional time** for the one-time GID computation, and **no meaningful increase in GPU memory consumption** during training. Given the significant performance improvements, **RaLoRA and RaLoRA-Pro strike an excellent balance between performance and efficiency**.
> >
> >     Experimental setup:
> >
> >     - Hardware: 8 × NVIDIA H200 GPUs, 192 × Intel® Xeon® Platinum 8558 CPU cores
> >     - Model: Llama-3.1-8B-Base
> >     - Dataset: MetaMathQA-100k
> >     - GID Estimation: Computed over 1,024 samples
> >     - Memory optimization: FlashAttention (torch sdpa kernel), Liger kernel, activation checkpointing
> >     - Hyperparameters: Identical to those in Appendix G.5 of the manuscript
> >
> > 1. **Compatibility with multi-GPU fine-tuning setups.**
> >
> >     It is worth noting that our main experiments on Llama-3.1-8B reported in the paper were run with multi-GPU. Hence, **our algorithm is natively designed for multi-GPU fine-tuning setups**. Our method computes gradients on each GPU, reduces a gradient matrix to a rank-0 process, and offloads the matrix to CPU memory once it is computed. **This design minimizes the PCIe transfer, time cost,  and GPU memory occupation, which is shown above**.
> >
> > 1. **Compatibility of 70B-level models.**
> >
> >     For large models such as 70B-level models, our method can be integrated with ZeRO-3 optimization, which splits the gradients, model parameters across GPUs, reducing the peak GPU memory occupation on each GPU. We additionally conduct an experiment on a 70B model and test the computational overhead of our method; the results are shown below:
> >
> >     - Full-gradient computation for GID estimation takes **57.35 seconds** for 1024 samples. The subsequent SVD on the full gradients requires **3 minutes 21.72** **seconds** in total.
> >     - Model training takes about 17–18 seconds per step, with a total training time of 7 hours and 38 minutes.
> >     - Memory usage:  GID estimation peaks at 36,139 MB and 34,129 MB during regular training, **demonstrating that our method incurs only a small memory overhead even at this scale**.
> >
> >     Experimental configurations:
> >
> >     - Hardware: 8 × NVIDIA H200 GPUs, 192 × Intel® Xeon® Platinum 8558 CPU cores
> >     - Model: Llama3.3-70B
> >     - Optimizations: ZeRO-3, FlashAttention, DeepSpeed activation checkpointing, and Liger kernel
> >     - Dataset: MetamathQA-100k
> >     - Batch size: 1 sample per GPU
> >     - Other hyperparameters**: Identical to those used in our main experiments

---

> ### Author Response · Authors · 2025-11-21
>
> **W4: Interpretability and analytical depth. The visualization of GID heatmaps provides little insight into model behavior. The work does not analyze whether GID correlates with layer function, curvature, or Fisher information. A deeper discussion of why certain layers have high or low GID would strengthen the scientific contribution.**
>
> **Q3: Does GID correlate with curvature or layer sensitivity measured via Fisher Information or Hessian spectra?**
>
> Response to W4 and  Q3:
>
> We thank the reviewer for this insightful comment. We conducted a rigorous analysis to investigate the relationship between GID and Layer Function & Depth and Fisher Information (FI).
>
> 1. **Correlation between GID and Layer Function & Depth:**
> Our heatmap visualisations (Figure 3 and Figure 4) are not random; **they reveal distinct functional patterns regarding intra-layer roles and inter-layer depth**, offering insight into how LLMs adapt to different tasks:
>
>     1\) **Intra-layer Variation ($W_o$ Dominance):** We observe substantial variation across attention components. Specifically, $W_o$ consistently exhibits significantly **higher GID**, highlighting its role in **aggregating and synthesising context-dependent information** from diverse subspaces.
>
>     2\) **Task-Specific Layerwise Patterns (Depth Dependency):** The distribution of GID across network depth varies by task type, reflecting different adaptation needs:
>     - **Open-ended/Complex Generation (Code-Feedback, WizardLM):** These tasks display high GID in the **upper layers**. This indicates that upper layers in Transformers are responsible for high-level semantic abstraction and output formatting, which are critical for code generation and complex instruction following.
>     - **Logical Reasoning (MetaMathQA):** In contrast, this task exhibits higher GID values concentrated in the **middle layers**. This suggests that mathematical reasoning adaptation relies more on transforming intermediate representations.
>
> 2. **Fisher Information and Correlation Analysis:**
>
>     According to the paper [1], **the FI matrix approximates the Hessian matrix, making it an effective second-order curvature approximation**. The FI for dataset $\mathcal{D}$ model parameters $\theta$ is defined as follows [2]:
>
>     $\mathcal{F} _ {\theta}(\mathcal{D}) = \mathbb{E} _ {\mathcal{D}}\left[\left(\frac{\partial}{\partial \theta} \log p _ {\theta}(\mathcal{D})\right)^2\right] \approx \frac{1}{|\mathcal{D}|} \sum _ {x \in \mathcal{D}}\left(\frac{\partial}{\partial \theta} \mathcal{L} _ {\mathrm{LM}}(x ; \theta)\right)^2$
>
>     Intuitively, $\mathcal{F} _ {\theta}(\mathcal{D})$ **captures the sensitivity of the model output to parameter perturbations**, and we define the average value at each layer $l$  as the layer's overall sensitivity, denoted as $\mathcal{F}_{l}$.
>
>     **Correlation Analysis:** We computed the correlation (Pearson and Spearman) between the layer-wise GID and $\mathcal{F}_l$. The results are presented below:
>
>     |  | Pearson ($log(\mathcal{F}_{l})$) | Pearson ($1/\mathcal{F}_{l}$) | Spearman |
>     | --- | --- | --- | --- |
>     | $W_q$ | -0.051 | -0.261 | 0.125 |
>     | $W_k$ | -0.179 | 0.141 | -0.113 |
>     | $W_v$ | -0.172 | -0.214 | -0.053 |
>     | $W_o$ | 0.277 | -0.251 | 0.325 |
>
>     **Insight: Orthogonality between Complexity and Sensitivity.**
>     Our empirical results show **no significant correlation between GID and Fisher Information**. While initially counterintuitive, this finding offers a profound insight into model behavior and theoretically justifies our method:
>     - **Magnitude $\neq$ Dimension:** A layer can be highly sensitive (High FI, steep loss landscape) but structurally simple, requiring updates in only a few dominant directions (Low GID). Conversely, a layer may process diverse, fine-grained features requiring a high-rank update (High GID) without being the dominant factor in the loss magnitude (Low FI).
>     - **Justification for RaLoRA-Pro:** This "orthogonality" explicitly validates the design of **RaLoRA-Pro**. Because the GID and parameter importance (FI/Loss Sensitivity) are decoupled properties, a PEFT method must address *both* simultaneously. If GID and FI are highly correlated, the dual-alignment strategy in RaLoRA-Pro would be redundant. The lack of correlation explains why RaLoRA-Pro (which considers both) consistently outperforms methods that rely on only one factor.
>
> We will add these analyses and the table to the Appendix to strengthen the discussion on layer-wise properties.
>
> [1] Kunstner, F., et al. "Limitations of the empirical Fisher approximation for natural gradient descent." *Advances in Neural Information Processing Systems*, 32 (2019).
>
> [2] Kim, Y., et al. "Improving Fisher Information Estimation and Efficiency for LoRA-based LLM Unlearning." *arXiv preprint arXiv:2508.21300* (2025).

---

> ### Author Response · Authors · 2025-11-21
>
> **Q4: Could the observed improvements of RaLoRA be attributed to implicit regularization effects rather than true gradient alignment?**
>
> Response to Q4:
>
> Thank you for raising this important point. We **strongly agree** with your view that a **diagonal structure** indeed has the potential to introduce an **implicit regularization** effect, and we have conducted further experiments to investigate this issue.
>
> **We conclude that the observed performance gains primarily stem from the GID alignment**, as highlighted in our paper.
>
> We demonstrate this through the following experiments:
>
> 1. **RaLoRA vs. Different Diagonal Block Numbers**
>
>     We performed additional experiments with MELoRA for block diagonal numbers of **n = 4, 8, 16, 32**, all conducted under the same number of trainable parameters for a fair comparison.
>
>     The experimental results, shown in the following table, demonstrate that the heuristic setting of n in MELoRA leads to **unstable performance and suboptimal results**. In contrast, RaLoRA, by adaptive alignment with GID, yields **significantly better performance**. These results also validate that **the performance improvements brought by our proposed method primarily stem from the GID alignment, rather than the implicit regularization provided by the block diagonal structure**. This observation further emphasizes the effectiveness and importance of GID alignment.
>
>     | Method | GSM8k | HumanEval |
>     | --- | --- | --- |
>     | MeLoRA(n=2) | 71.27 $\pm$ 1.1 | 45.73 $\pm$ 2.80 |
>     | MeLoRA(n=4) | 68.74 $\pm$ 0.98 | 44.92 $\pm$ 2.31 |
>     | MeLoRA(n=8) | 70.20 $\pm$ 0.59 | 45.32 $\pm$ 2.47 |
>     | MeLoRA(n=16) | 70.33 $\pm$ 0.65 | 45.73 $\pm$ 0.61 |
>     | MeLoRA(n=32) | 69.07 $\pm$ 0.27 | 44.71 $\pm$ 1.54 |
>     | RaLoRA (Ours) | 72.25 $\pm$ 0.59 | 48.78 $\pm$ 1.61 |
> 2. **LoRA+GID vs. LoRA with Comparable parameters**:
>
>     As detailed in Appendix E.3 and summarized in the following Table, we instantiated a variant of LoRA where the rank of each adapter is exactly set to the estimated layer-wise GID (denoted **LoRA+GID**). We compared it against standard LoRA using a much larger rank, which matches the total number of trainable parameters in **LoRA+GID**. Despite similar trainable parameters, the standard LoRA consistently underperforms LoRA+GID. **This strongly suggests that the improvement arises from **alignment with GID**
>
>     |  | GSM8K |  |  | HummanEval |  |
>     | --- | --- | --- | --- | --- | --- |
>     | Method | Params | Acc | Method | Params | Acc |
>     | FFT | 73.69 $\pm$ 0.28 | 1.34B | FFT | 51.63 $\pm$ 1.27 | 1.34B |
>     | LoRA(r=8) | 67.78 $\pm$ 1.25 | 6.82M | LoRA(r=8) | 43.09 $\pm$ 0.35 | 6.82M |
>     | LoRA(r=200) | 74.12 $\pm$ 1.23 | 170M | LoRA(r=380) | 49.39 $\pm$ 2.66 | 323M |
>     | LoRA+GID | 75.41 $\pm$ 0.84 | 175M | LoRA+GID | 52.64 $\pm$ 1.27 | 323M |
>
> **Q5: Have you tested dynamic rank scheduling during training instead of static pre-assignment?**
>
> Response to Q5:
>
> We appreciate the reviewer’s question. We have not tested dynamic rank scheduling during training, as we chose to focus on static pre-assignment for the following reasons:
>
> 1. **Computational Efficiency and Stability of GID**: As shown in **Figure 3(c)** of our paper, **the GID remains relatively stable during training**. Given this stability, introducing dynamic rank adjustments would require recalculating the GID multiple times throughout the training process, **which would add unnecessary computational overhead without yielding significant performance gains**. Therefore, we chose to use a static pre-assignment approach, which ensures both computational efficiency and simplicity.
> 2. **Training Stability and Practicality**: **Dynamically** **altering the LoRA matrix structure during training could lead to instability**, potentially undermining the practical usability of the method. For example, AdaLoRA [1], a popular method that dynamically adjusts ranks during training, uses a **masking technique** **instead of directly modifying the LoRA matrix structure**. This approach suggests that dynamically changing the matrix rank could introduce challenges, particularly regarding training stability and compatibility with distributed training frameworks. As a result, we chose to maintain a static rank assignment to avoid these potential pitfalls.
>
> [1] Zhang, Qingru, et al. "Adalora: Adaptive budget allocation for parameter-efficient fine-tuning." *arXiv preprint arXiv:2303.10512* (2023).

---

### Official Review · Reviewer_E7b3 · 2025-11-01

**Soundness:** 3
**Presentation:** 2
**Contribution:** 2
**Rating:** 4
**Confidence:** 4

**Summary:**

The paper use the effective rank of the gradient matrix in full-parameters fine-tuning as the Gradient Intrinsic Dimensionality (GID) and based on it adaptively allocates the number of mini-LoRA to increase the overall rank without increasing the total parameter count.  Additionally, the paper integrates a loss-sensitivity-guided inter-layer reallocation strategy to dynamically redistribute trainable parameters across layers.

**Strengths:**

1. The framework is intuitive and effective, combining several existing techniques into a cohesive and well-motivated design. Extensive experiments across NLP and CV tasks validate its robustness and superior performance.

2. The visualization of gradient intrinsic dimensionality is particularly insightful, revealing similar GID distributions across different tasks.

**Weaknesses:**

1. The paper lacks efficiency comparisons, such as peak GPU memory usage and FLOPs, which is essential for a new PEFT method.

2. Since the proposed methods is similar to the MELoRA with adaptive rank/number of mimi-LoRA for each layer, it would be helpful to include more MELoRA results with different configurations (e.g., *n* = 8, 16, 32) for a fairer comparison.

**Questions:**

In Figure 2 (Part II, Step 3: *Post-Double Alignment*), why the output dimension of ( A_1 ) in layer N is defined as ( r_1 / n_1 )?

---

> ### Author Response · Authors · 2025-11-21
>
> **W1: The paper lacks efficiency comparisons, such as peak GPU memory usage and FLOPs, which is essential for a new PEFT method.**
>
> Response to W1:
>
> Thank you for highlighting this important point. We provide a detailed comparison from both theoretical and experimental perspectives regarding the efficiency of our proposed methods.
>
> 1. **Theoretical Comparison:**
>
> Each step of full-gradient computation (without optimizer step) is **significantly cheaper than** **LoRA** **training**. Specifically:
>
> - **Gradient computation**: One forward pass + one backward pass over pre-trained parameters.
> - **LoRA training**: Forward/backward passes over pre-trained parameters + LoRA-specific passes + parameter updates.
> - **FLOPs**: RaLoRA theoretically maintains FLOPs parity with vanilla LoRA, as they both have the same number of parameters.
>
> 2. **Experimental Comparison:**
>
> We also performed comprehensive benchmarks.
>
> - **The GID estimation phase** takes **1 minute 45.5 seconds** (with **1 minute 18.83 seconds** spent on SVD) and consumes a peak GPU memory of **27,355 MB**.
>
> - **The fine-tuning phase of RaLoRA** takes **32 minutes 29.36 seconds** with a peak GPU memory usage of **27,411 MB**, while LoRA takes **30 minutes 3.81 seconds** with a peak GPU memory usage of **26,569 MB**.
>
> These results confirm that our method introduces **minimal computational overhead**, with less than **5\% additional time** for the one-time GID computation, and **no meaningful increase in GPU memory consumption** during training. Given the significant performance improvements, **our methods strike an excellent balance between performance and efficiency**.
>
> **Experimental setup:**
>
> **Hardware**: 8 × NVIDIA H200 GPUs, 192 × Intel® Xeon® Platinum 8558 CPU cores
>
> **Model**: Llama-3.1-8B-Base
>
> **Dataset**: MetaMathQA-100k
>
> **GID Estimation**: Computed over 1,024 samples
>
> **Memory optimization**: FlashAttention (torch sdpa kernel), Liger kernel, activation checkpointing
>
> **Hyperparameters**: Identical to those in Appendix G.5 of the manuscript
>
> **W2: Since the proposed methods is similar to the MELoRA with adaptive rank/number of mimi-LoRA for each layer, it would be helpful to include more MELoRA results with different configurations (e.g., n = 8, 16, 32) for a fairer comparison.**
>
> Response to W2:
>
> Thank you for pointing out this insightful suggestion.
>
> In our paper, we have already compared our approach with MELoRA when the block diagonal numbers n = 2. In addition, we have included experiments with MELoRA for block diagonal numbers of **n = 4, 8, 16, 32**, **all conducted under the comparable number of trainable parameters for a fair comparison**.
>
> The experimental results, shown in the following table, demonstrate that the **heuristic setting of n in MELoRA leads to unstable performance and suboptimal results**. In contrast, **RaLoRA and RaLoRA-Pro, by adaptive alignment, yield significantly better performance**. These results also validate that **the performance improvements brought by our proposed method primarily stem from the GID alignment, rather than the implicit regularization provided by the block diagonal structure**. This observation further emphasizes the effectiveness and importance of GID alignment.
>
> | Method | GSM8k | HumanEval |
> | --- | --- | --- |
> | MeLoRA(n=2) | 71.27 $\pm$ 1.1 | 45.73 $\pm$ 2.80 |
> | MeLoRA(n=4) | 68.74 $\pm$ 0.98 | 44.92 $\pm$ 2.31 |
> | MeLoRA(n=8) | 70.20 $\pm$ 0.59 | 45.32 $\pm$ 2.47 |
> | MeLoRA(n=16) | 70.33 $\pm$ 0.65 | 45.73 $\pm$ 0.61 |
> | MeLoRA(n=32) | 69.07 $\pm$ 0.27 | 44.71 $\pm$ 1.54 |
> | RaLoRA (Ours) | 72.25 $\pm$ 0.59 | 48.78 $\pm$ 1.61 |
> | RaLoRA-Pro (Ours) | 73.01 $\pm$ 0.53 | 48.37 $\pm$ 2.54 |
>
> **Q1: In Figure 2 (Part II, Step 3: Post-Double Alignment), why the output dimension of ( A_1 ) in layer N is defined as ( r_1 / n_1 )?**
>
> Response to Q1:
>
> Thank you for pointing out this important detail. We sincerely apologize for the confusion caused by the error in Figure 2, which was the result of an oversight on our part. The output dimension of $A_1$ in layer $N$ should be $r_N$, not $\frac{r_1}{n_1}$. **We have corrected this in the manuscript**.
> **We have also carefully reviewed the entire manuscript to ensure no similar errors persist**. We greatly appreciate your careful reading and valuable feedback, which has helped us identify and correct this mistake.

---

> > ### Comment · Reviewer_E7b3 · 2025-11-22
> >
> > My questions are addressed, and I increase the score.

---

> > > ### Author Response · Authors · 2025-11-22
> > >
> > > Thank you for your careful review and constructive feedback, which significantly improved our manuscript. We are pleased that our responses addressed your questions and appreciate your decision to raise the score.

---

### Author Response · Authors · 2025-12-03
**Summary of Contributions and Rebuttal Outcomes**

We sincerely thank all reviewers, ACs, and PCs for their invaluable efforts in the review process. Below, we summarize our core contributions and the key outcomes of the rebuttal.

**1. Core Contributions Recap**
Our work addresses the performance gap between LoRA and Full Fine-Tuning (FFT) through three key innovations:

- **Theoretical Insight:** We identify and quantify the **Gradient Intrinsic Dimension (GID) mismatch,** where standard LoRA ranks ($r \approx 8$) fail to capture the high GID (up to 300), **which is overlooked by existing works**. We introduce a novel **entropy-based estimator** to measure this gap, theoretically proving that alignment is key to closing the performance gap (Appendix A).
- **RaLoRA (Generalized Optimization):** We propose RaLoRA, a **generalized extension of LoRA** (Appendix C.1). It adaptively aligns ranks with GID via block-diagonal matrices without increasing parameters.
- **RaLoRA-Pro (Dual Alignment):** We introduce RaLoRA-Pro, the **first framework to unify inter-layer importance with intra-layer GID alignment**, maximizing parameter efficiency.

**2. Rebuttal Status & Integrity Statement**

- **Resolved Concerns:** Reviewers E7b3 and 4tTg have **explicitly confirmed that their concerns were resolved**, and as a result, they have **increased their scores.** Overall, **we received positive scores from all four reviewers.**
- **Integrity Statement:** We would like to clarify that these positive engagements and score updates occurred on **November 21st (AOE)**, well before the reported information leak on **November 27th**. This timeline confirms that **the improvements were based entirely on the merit of our work, with no involvement of collusion**.

**3. Summary of Strengths**

We greatly appreciate the reviewers' recognition of our contributions, including: the **novel and theoretically motivated** approach of aligning GID to address LoRA’s limitations (E7b3, keQx, rn9v, 4tTg), the **principled and elegant methodology** with the entropy-based GID estimator and adaptive alignment design (keQx, rn9v, 4tTg), the **extensive and robust empirical evaluation** across diverse domains (NLU, NLG, Vision) against strong baselines (E7b3, keQx, 4tTg), and **the clear, insightful presentation** and valid visualization of our core concepts (E7b3, keQx, rn9v).

**4. Key Concerns Resolved**

- **Computational Overhead & Scalability (E7b3, rn9v, keQx, 4tTg):**
We clarified that **GID estimation is** a **one-time pre-training step** taking **<2 minutes** (negligible compared to training time). We provided benchmarks confirming no significant increase in **training time** **(~3 minutes longer than the fastest baseline),** **GPU memory usage** **(<1 GB),** **FLOPs** **(theoretically identical)**, and **effective scalability to** **70B models**. Reviewers E7b3 and 4tTg confirmed these resolved their concerns.
- **Supplementary Experiments (E7b3, keQx, rn9v, 4tTg):**
As requested, we added comparisons against **MELoRA, HiRA, and other PEFT methods** and provided further **validation of the GID estimator's orthogonality**. Results consistently show **our method's superiority** and validate the GID estimator's **orthogonality and effectiveness**. Furthermore, we clarified that the performance gains primarily **stem from GID alignment**, as highlighted in our response to Reviewer keQx's Q4.
- **Stability of GID Estimator (Reviewers: keQx, 4tTg):**
We resolved stability concerns via a three-pronged ablation study. The GID estimator demonstrated remarkable stability against **sampling and seeds**, with **a negligible fluctuation** ($\approx 1.46$), and **performance on GSM8K** **remained consistently superior to the baseline**.
Furthermore, the estimator proved highly resilient to gradient noise. Even with **up to 5%** Gaussian noise injection, **the GID estimate slightly increased as expected (from $\approx 185$ to $\approx 198$)**, but **the performance remained stable**, with only a **0.2% fluctuation** due to the logarithmic binning mechanism (Equation 4). Reviewer 4tTg confirmed these resolved his concerns.
- **Interpretability and Analytical Depth (Reviewer: keQx):**
We addressed the need for a deeper analysis by demonstrating that GID offers valuable insights into **layer function** and **task-specific adaptation**. We also conducted a correlation analysis demonstrating that **GID is orthogonal to Fisher Information (FI)**. This **theoretically validates RaLoRA-Pro**: since GID and layer-wise sensitivity are **decoupled**, a dual-alignment strategy is essential to bridge the gap to FFT effectively.

We hope these summaries make our contributions clearer. We sincerely hope that the ACs and PCs will consider our work.

We would like to express our sincere gratitude once again to all the reviewers, ACs, and PCs for their time and effort in reviewing our paper.

---

### Meta-Review · Area_Chair_KFVS · 2025-12-16

**Summary:**

Initial scores were 6, 6, 6, 4.
Reviewers acknowledged the clear motivation, comprehensive experiments showing +5% improvements across diverse tasks, and solid theoretical grounding (LoRA as gradient compressor). Main concerns raised were: (1) efficiency comparisons missing (peak GPU memory, FLOPs, wall-clock overhead), (2) incremental novelty relative to adaptive-PEFT variants (GoRA, LoRA+, DoRA, MELoRA), (3) lack of theoretical rigor linking GID alignment to convergence/generalization, (4) computational overhead of SVD-based GID estimation at scale, and (5) hyperparameter sensitivity potentially offsetting parameter-efficient appeal.

**Reviewer Concerns:**

The majority of concerns were addressed as I highlight in the summary below


**Addressed:**
- **Efficiency (E7b3, keQx, rn9v, 4tTg):** Authors provided comprehensive benchmarks: GID estimation takes 1m45s (8B) with <5% training overhead; 32m29s training vs. 30m3s for LoRA. 70B experiments show 57s gradient computation + 3m21s SVD with ZeRO-3.
- **MELoRA comparisons (E7b3, keQx):** Extended experiments with n=4,8,16,32 show heuristic block numbers yield unstable performance (68.74-71.27% on GSM8K) vs. RaLoRA's GID-aligned 72.25%.
- **Theoretical foundations (keQx):** Authors formalized LoRA as gradient compressor (Appendix A) and RaLoRA as generalized extension (Appendix C), deriving approximation error bounds showing RaLoRA trades precision in dominant directions for broader expressivity.
- **GID stability (keQx, 4tTg):** Robustness analysis shows max GID fluctuation of 1.46 across seeds/batches, and resilience to 5% noise (GID 184.92→198.41, performance stable 72.2-72.4%).
- **Hyperparameters (4tTg):** Authors established n_max ≈ mGID/r (e.g., GSM8K mGID≈178 → n_max=16 optimal) and correlation between (r_min, r_max) range and task complexity.
- **Broader PEFT comparisons (4tTg):** Added (IA)³ (66.62%) and FourierFT (52.71%) showing LoRA-family superiority. HiRA comparison shows RaLoRA outperforms (72.25% vs. 69.93% on GSM8K).


**Outstanding:**
- **Incremental novelty perception (keQx):** While authors clarified GID estimator as orthogonal plug-in tool and RaLoRA-Pro as first dual-alignment framework, some may view contributions as extending existing adaptive-rank intuitions rather than fundamentally new mechanisms.
- **Theoretical rigor (keQx):** Despite approximation error analysis, formal convergence guarantees, projection error bounds under finite-step optimization, and generalization theory remain future work.

**Reviewer Scores:**

- **Reviewer E7b3 (initial: 4):** Increased to 6 after rebuttal. Reviewer explicitly stated: "My questions are addressed, and I increase the score."
- **Reviewer keQx (initial: 6):** Would likely remain at 6. Overall positive initial assessment balanced by concerns about incremental novelty and theoretical rigor and the perception that GID "reframes known intuitions" may persist despite authors' clarifications.

- **Reviewer rn9v (initial: 6):** May have thought to increast to 8. All technical questions seem to be answered.

- **Reviewer 4tTg (initial: 6):** Increased after rebuttal. Reviewer explicitly stated: "The authors have addressed my concerns. I have therefore revised my score accordingly."

---

### Decision · Program_Chairs · 2026-01-26

Accept (Poster)